# Bridging the Translational Gap: Rethinking Smooth Muscle Cell Plasticity in Atherosclerosis Through Human-Relevant In Vitro Models

**DOI:** 10.3390/cells14231913

**Published:** 2025-12-02

**Authors:** Liliana Som, Nicola Smart

**Affiliations:** Department of Physiology, Anatomy and Genetics, Institute of Developmental & Regenerative Medicine, BHF Oxford Centre of Research Excellence, University of Oxford, Oxford OX3 7TY, UK; liliana.som@exeter.ox.ac.uk

**Keywords:** smooth muscle cell, phenotypic plasticity, in vitro modelling, human-relevant, omics

## Abstract

**Highlights:**

**What are the main findings?**

**What are the implications of the main findings?**

**Abstract:**

While vascular smooth muscle cell (SMC) plasticity is increasingly recognised as a critical driver of atherosclerosis progression, most mechanistic insights derive from murine models that fail to fully capture the diversity and complexity of human SMC phenotypes. This creates a translational gap in our understanding of disease-relevant cell states. Human single-cell and genetic studies reveal a broader spectrum of SMC phenotypes, many of which remain uncaptured by existing experimental models. In this review, we argue that better human in vitro models, when critically assessed and integrated with omics data from human disease, can help bridge this gap. We examine how different in vitro systems, from simple monocultures to advanced co-culture and 3D platforms, can model human SMC plasticity, and how benchmarking against human single-cell and multi-omics data can guide model selection, validation, and refinement.

## 1. Introduction

Cardiovascular diseases (CVDs) are the leading cause of morbidity and mortality worldwide. Atherosclerosis is a chronic disease of the arterial vessel wall, whereby, in response to endothelial injury, atheromatous plaques form, calcify, constrict, and occlude the lumina of medium and large arteries. Atherosclerotic plaques progress through well-defined pathological stages, with adaptive intimal thickening caused by SMC infiltration followed by lipid accumulation in the core causing pathologic intimal thickening. With disease progression, a fibroatheroma is formed with an inflammatory necrotic core under an SMC-populated, extracellular matrix (ECM)-rich fibrous cap. In advanced disease stages, calcification results in the formation of a fibrocalcific plaque [1,2,3]. Plaque rupture has severe clinical consequences, leading to interruption of blood flow to affected areas, causing myocardial infarction, ischemic stroke, and peripheral arterial disease [3,4].

During the advancement of plaque formation, various cell types accumulate in the lesion, such as inflammatory and immune cells, lipid-laden foamy macrophages, endothelial cells (ECs), and both native and phenotypically modulated SMCs. Notably, SMCs are thought to represent the most abundant cell type in human atherosclerotic plaques, constituting as much as 90% of cells in some stages of lesion progression [5,6]. The progression of CVDs, such as atherosclerosis, depends on the response of SMCs in the medial layer of the vessel wall.

### 1.1. SMC Plasticity in Vascular Biology

SMCs are a crucial component of the vasculature, the most abundant cell type in the arterial vessel wall. In healthy vessels, fully differentiated SMCs are quiescent and appear in an elongated, spindle-shaped morphology in the media. They are responsible for regulating vascular tone, stabilising the vessel wall, maintaining ECM homeostasis, and coordinating vascular contraction and relaxation [7].

This differentiated state, however, is not irreversible. SMCs retain the ability to dedifferentiate to alternative phenotypes in response to environmental cues, a characteristic referred to as phenotypic plasticity. The phenotypic transition is an intrinsic property of SMCs, enabling them to respond to physiological and pathophysiological stimuli and actively participate in tissue remodelling and repair processes. The plasticity of SMCs is essential during embryonic development, where they contribute to the formation of the vascular system [8,9]. Switching between contractile and synthetic phenotypes is critical for tissue homeostasis and repair; however, dysregulation of this process can contribute to the progression of various diseases [10,11].

### 1.2. SMC Plasticity in Disease

In pathological conditions, during vascular remodelling following endothelial damage and dysfunction, SMCs undergo dedifferentiation and acquire high proliferative and migratory capabilities, and enhance extracellular matrix synthesis [12,13]. In vitro, the associated cytoskeletal rearrangements typically manifest as a shift from the spindle-shaped contractile morphology to a rhomboid, synthetic-like shape under culture conditions. Various studies demonstrate that phenotypically modulated SMCs can both positively and negatively contribute to disease progression, highlighting the diverse, context-dependent functions of SMCs [14,15]. In atherogenesis, synthetic SMC phenotypes play a key role in the progression of atherosclerotic lesion development and contribute various cell types to expand the necrotic core, enhancing inflammation, fibrosis, and calcification. Conversely, they are also essential for the formation of fibrous caps, stabilising the plaques and making them less prone to rupture. This duality highlights the importance of understanding SMC fate in plaque evolution, a key consideration for therapeutic intervention [15,16].

### 1.3. The Translational Gap from Mice to Humans

Inhibition of the SMC phenotypic switch has been shown to attenuate the progression of vascular disease in animal models [17,18,19]. While these studies in murine models have been invaluable in identifying key regulators of SMC phenotypic modulation, translating these findings to human disease remains challenging, hindering therapeutic development. Differences in vascular architecture, immune responses, and plaque composition between mice and humans limit the direct applicability of these insights to clinical settings.

### 1.4. Aims and Scope of This Review

To investigate the mechanisms that serve to protect the human vasculature and how they may be targeted to alleviate disease, there is a pressing need for studies in physiologically relevant human models. Recent advancements in the approval of in vitro models as an alternative for animal testing in drug development and biological applications were made by the FDA Modernisation Act 2.0 in 2022 [20,21,22]. This shift underscores the need for physiologically relevant alternatives to animal models that more accurately recapitulate human biology. Establishing a more accurate in vitro model of human arteries in health and disease will allow the interrogation of molecular mechanisms and the identification of potential genetic and pharmacological targets for CVD. A disease-relevant co-culture system would enable the investigation of the effects of SMC switching on EC-SMC crosstalk, advancing our understanding of coronary artery disease progression, and facilitating the development of effective therapies.

Given that SMC phenotypes and functions vary significantly across vascular beds, this review focuses specifically on arterial SMCs. Venous SMC biology, including differences in developmental origin, ECM organisation, and mechanical loading, is beyond the scope of this review. Therefore, all references to SMCs refer to arterial SMCs unless otherwise specified.

The recent comprehensive review by Azar et al. explores human SMC plasticity, spatial localisation and emerging imaging or therapeutic targeting perspectives [23]. Our review extends this by critically evaluating human in vitro model systems and benchmarking them against single-cell, spatial, and genetic datasets, with the aim of closing the translational gap.

## 2. SMC Phenotypic Diversity in Atherosclerosis

### 2.1. From a Binary View to a Spectrum of States

Historically, SMC plasticity was simplistically viewed as a binary transition between two distinct phenotypic states, the contractile and synthetic phenotypes. In healthy vessel walls, the contractile, quiescent and spindle-shaped SMCs express high levels of contractile proteins, such as α-smooth muscle actin (encoded by *ACTA2*), transgelin (*TAGLN*), calponin (*CNN1*), and smooth muscle myosin heavy chain (*MYH11*), supporting vascular tone and stability. The identification of these classic contractile markers provided essential tools for studying SMC identity and contributed to the establishment of the binary contractile-synthetic paradigm [24,25,26,27].

The binary classification emerged from in vitro studies in primary SMCs that were exposed to growth factors, such as platelet-derived growth factor B (PDGF-BB). In these early studies, the dedifferentiation in vitro was defined by loss of the established contractile markers and an increase in proliferation, migration, and enhanced ECM production—hallmark characteristics of the synthetic phenotype [28,29,30]. This dichotomous view has provided invaluable groundwork for understanding vascular remodelling in the context of CVDs, such as atherosclerosis, restenosis and aneurysm formation.

However, this directional transitional view has proven to be overly simplistic, in light of the advancements arising from lineage tracing studies and high-resolution tools, especially single-cell transcriptomics and fate mapping. With these technologies, a diverse range of intermediate and transdifferentiated states has emerged. A growing body of evidence supports the idea that the SMC phenotypic switch is a reversible, dynamic transition along a spectrum of phenotypic states across the SMC continuum, in response to environmental cues, mechanical insults, and inflammatory stimuli [31,32,33]. This shift in understanding of SMC biology enables a deeper exploration and comprehension of SMC behaviour in health and disease.

### 2.2. Expanding the Spectrum: Emerging SMC Phenotypes

Lineage-tracing studies and single-cell transcriptomics have identified distinct SMC states, including Myofibroblast-like, Mesenchymal-like, Macrophage-like, Osteoblast-like, Chondrogenic-like, and Adipocyte-like populations (Figure 1) [32,34]. These states differ in their gene expression profiles, the stimuli that drive their dedifferentiation, functional properties, and their roles in atherosclerotic plaque development. The switch is characterised by the transformation of the contractile apparatus, loss of contractile markers and alteration of cellular behaviour, such as proliferative and migratory capacities, often involving ECM remodelling and exacerbation of inflammation. The canonical contractile phenotype-associated markers include *ACTA2*, *MYH11*, *TAGLN*, and *CNN1*. However, it is important to note that some of these are not exclusively expressed in SMCs. Some makers indicate synthetic phenotypes, such as metalloproteinase-9 (MMP9) and osteopontin (OPN), but the list of these is less well-established and lacks universally accepted exclusive markers that do not overlap with other cell lineages [35].

#### 2.2.1. Insights from Murine Lineage Tracing

Utilisation of transgenic mice for lineage tracing studies has provided substantial evidence of SMC phenotypic switching occurring in vivo. The *Tagln/SM22α*-Cre (constitutive) and *Tagln*-CreERT2 and *Myh11*-CreERT2 (tamoxifen-inducible) mouse lines have been widely used to fate map SMC populations. Early fate-mapping studies employed *Tagln*-CreERT2 mice to achieve inducible recombination in SMCs, enabling fate mapping with high affinity. The foundational work by the Feil lab in creating both transgenic and knock-in *Tagln*-CreERT2 lines established the tools for inducible SMC fate mapping [36]. Early disease-model application of *Tagln*-CreERT2 lineage demonstrated SM22α^+^ SMC-derived cells traced within neointimal lesions in atherosclerosis [37].

Innovations such as localised tamoxifen delivery and adult-specific recombination protocols further expanded the model’s versatility [38,39]. However, specificity concerns have emerged due to the low specificity of SM22α expression for mature SMCs, particularly in disease settings, and high presence in non-SMC lineages during development and under inflammatory conditions, which can result in the labelling of non-SMC populations, especially with the use of constitutive Cre lines [25,40]. In contrast, *Myh11*-CreERT2 is considered a more specific marker of differentiated, contractile SMCs and is typically preferred for studies aiming to trace mature SMC fate with greater fidelity. This difference in specificity can affect the interpretation of phenotypic transitions, leading to greater reliance on *Myh11*-CreERT2 for tracing mature medial SMCs [41]. Together, these lineage tracing models have revealed a wide spectrum of SMC-derived phenotypes in mice; however, interpretation is constrained by the reliance on Cre drivers with variable specificity. Moreover, it is important to recognise that murine models of atherosclerosis do not develop arterial disease spontaneously and depend on severe genetic (e.g., *ApoE*^-/-^ or *Ldlr*^-/-^ backgrounds) or dietary modifications (e.g., high-fat or high-cholesterol diet). Consequently, insights gained from these models may only reflect certain aspects of human SMC behaviour and should be interpreted with due caution.

Pioneering studies using *Myh11*-CreERT2 lines have identified SMC populations with diminished contractile marker expression and acquired properties associated with synthetic phenotypes. For example, Shankman et al. showed the emergence of macrophage-like SMCs that express galectin-3 (*Lgals3*) and CD68 [10]. Furthermore, Wirka et al. demonstrated the differentiation to a fibromyocyte population, expressing fibroblast markers such as *Lum*, *Dcn* and *Bgn* [18]. Pan et al. described an intermediary state during the transition, termed SEM (stem cells, endothelial cells, monocytes) [42]. These fate-mapped populations were confirmed to originate from SMCs, despite their transcriptomic profiling resembling other lineages. Consistent with these findings, Wang et al. concluded that SMCs contribute the majority of foam cells in *ApoE^-/-^* mice, which spontaneously develop hypercholesterolemia and advanced atherosclerotic lesions, representing one of the most widely used murine models of atherosclerosis. These findings are consistent with the proposal that macrophage-like and lipid-laden phenotypes can emerge from a SMC origin [43]; however, the derivation of cells with a bona fide macrophage (*CD45+*) phenotype is still strongly contested, with the suggestion that expression of markers, such as *Lgals3*, may simply denote SMCs with the capacity to phagocytose [44,45]. Collectively, lineage tracing studies highlight the remarkable diversity of SMC-derived cell states in murine atherosclerosis, as reviewed comprehensively by Bentzon and Majesky [46].

#### 2.2.2. Transcriptomic Insights into Human Atherosclerotic Lesions

Murine lineage tracing studies have provided detailed insights into SMC fate transitions. In human studies, the absence of fate-mapping tools means we rely on transcriptional signatures to infer the diversity of SMC plasticity, using single-cell RNA sequencing (scRNA-seq).

Multiple large-scale datasets of human plaques guide our understanding of SMC heterogeneity in atherosclerosis through the creation of transcriptomic atlases of the SMC-like populations. Wirka et al. confirmed that the ECM-remodelling, fibroblast-like state—the fibromyocyte population observed in the murine lineage tracing studies is present in advanced human coronary plaques as well, characterised by high fibronectin (*FN1*) and osteoprotegerin (*TNFRSF11B*) expression [18]. Pan et al. reported SMC-like populations enriched for inflammatory, myofibroblast-like and ECM-secreting transcriptional signatures in human coronary atherosclerotic plaques [42]. Alencar et al. identified an *LGALS3+*, ECM-remodelling pioneer population that gives rise to multiple SMC phenotypes, including an osteogenic phenotype characterised by upregulated *RUNX2* and mineralisation-gene expression in human carotid endarterectomy samples, contributing to vascular calcification and plaque destabilisation [47]. The scRNA-seq and scATAC-seq (Single cell assay for transposase-accessible chromatin using sequencing) of human carotid atherosclerotic plaques by Depuydt et al. generated an atlas of SMC phenotypes, confirming the cellular plasticity in human atherosclerosis. These sequencing methods resolved a contractile SMC cluster, expressing the canonical contractility markers (*MYH11*, *ACTA2*, *TAGLN*), a synthetic cluster with reduced contractile markers and enhanced ECM genes (*COL1A1*, *MGP*, *COL3A1*), a small cluster of *ACTA2*+ macrophage-like cells and an EC cluster co-expressing SMC markers, likely undergoing EndMT [48].

Whilst consistency across different species is observed, such as the robust detection of fibroblast-like and myofibroblast states, there are definite areas where disagreement on the existence of phenotypes, such as the macrophage-like cluster, highlights a translational tension. Although these datasets are critical for progressing our understanding of human SMC phenotypic heterogeneity, the limitation of spatial and temporal resolution in human samples offers an incomplete view of SMC plasticity, limiting definitive conclusions about cellular origins and transitions. It is important to acknowledge that dissociation-based single-cell techniques utilised in most of these studies may underrepresent small, fragile or transitional SMC-derived populations, due to the technical challenges in sample processing. Nevertheless, emerging datasets have begun to reveal overlapping and at times diverging phenotypic programmes across species.

### 2.3. The Translational Divide: Comparing Murine and Human Phenotypes

Inherent limitations in both murine lineage tracing studies and human transcriptomic studies are acutely apparent when SMC phenotypes are compared between species. Contrasting murine lineage-tracing studies with human single-cell atlases reveals both convergence and divergence in the spectrum of SMC phenotypes (Table 1).

Whilst the canonical SMC markers, such as *Tagln*, *Myh11* and *Acta2*, established from murine studies, are widely accepted to reflect the contractile phenotypic state, they do not always directly map onto human datasets. Due to differences in baseline expression and detection thresholds, often limited by the origin of human disease severity, sole reliance on these markers might lead to discrepancies in cell state definitions [35,49].

Several SMC phenotypes established in murine models have been observed or inferred in human atherosclerosis data as well, although their frequency and functional relevance may differ between species. Conserved phenotypes include ECM-producing, fibroblast-like SMCs, sharing conserved gene signatures such as enhanced expression of *LUM*, *FN1*, *COL1A1*, *DCN* and *BGN* [18,48,50]. Inflammatory signatures have been observed in both, although to varying degrees—less pronounced in human plaques, and often marked by distinct sets of inflammatory effectors, including cytokines and chemokines, with divergent expression profiles [18,48,51,52].

Discrepancies are most prominent in the reporting of macrophage-like SMCs and transitional states in human and murine plaques. Whilst the existence of macrophage-like clusters is widely reported in murine models, data from human plaques are rare and inconclusive [10,18,48,50,53,54]. Notably, in human coronary plaques, Wirka et al. identified a prominent fibromyocyte population as the major SMC-derived state but did not detect a discrete macrophage-like SMC cluster, underscoring dataset- and method-dependent detectability [18]. Conversely, in human carotid plaques, Depuydt et al. reported a small *ACTA2*^+^, *CD68*^+^ cluster with macrophage-like features, though its lineage identity remains uncertain [48]. Integrative meta-analysis across human single-cell studies recovers disease-relevant inflammatory/macrophage-like modules within SMC-lineage-adjacent states, although lineage ambiguity with myeloid cells remains [50].

The apparent scarcity of a macrophage-like SMC cluster in human plaques may be attributed to detection method limitations and a lack of clarity regarding the origin of cells. Further complicated by the lack of marker specificity between SMC-derived macrophages and true myeloid cells, this presents a challenge to distinguish discrete populations, especially in late-stage human samples, where contractile identity might be lost entirely [42]. Whilst fate-mapping in murine studies has inherent limitations, transitional SMC states are nonetheless more readily captured in these models, as human samples are typically derived from late-stage disease where intermediate phenotypes are no longer detectable [42,48].

Several phenotypes that have been long established in murine models are being recognised in human datasets, such as the Osteogenic-like and adipocyte-like [46,54,55]. Similarly, the mesenchymal-like state is well-characterised in murine models, but only identified in some integrative human datasets [31,48,50]. These differences underscore both the strengths and the limitations of murine models for capturing the full spectrum of SMC plasticity in human atherosclerosis.

Species-specific vascular architecture, plaque composition, and immune cell microenvironment are all key factors contributing to the modulation of SMC behaviour. Differences in the structural composition of the vascular set-up between species may be a fundamental source of the observed discrepancies in SMC function. Whilst in murine models, early-stage plaques can be readily harvested, access to human atherosclerotic plaques is often limited to late stages. Additionally, the technical limitations of human plaque analysis put a constraint on human-relevant discoveries. Furthermore, patient-to-patient heterogeneity, comorbidities, tissue availability and the lack of healthy controls further complicate the interpretation of these datasets [23,48].

Together, these differences underscore the need to develop experimental models that both capture the wide array of SMC phenotypes and, crucially, are benchmarked directly against human disease-derived phenotypic signatures, a prerequisite for closing the gap between animal and human studies, and more importantly, the gap between human studies and clinical intervention.

**Table 1 cells-14-01913-t001:** Overview of smooth muscle cell (SMC) phenotypes in animal models and humans. Summary of SMC phenotypic states characterised in atherosclerosis, outlining key molecular markers, experimental evidence from lineage-tracing and single-cell studies, and representative literature. The table contrasts conserved phenotypes, such as contractile and fibroblast-like SMCs, with less consistently defined states, including macrophage-like, adipocyte-like, and EndMT-adjacent cells, whose presence in human lesions remains debated.

Phenotype/State	Key Markers/Signatures	Evidence in Animal Models	Evidence in Humans	Representative Studies	Notes/Caveats
Contractile (baseline)	*ACTA2*, *MYH11*, *TAGLN*, *CNN1*	Robustly detected in healthy vessels and lineage-traced SMCs in *ApoE*^−/−^ mice; downregulated under atherogenic conditions.	Rarely preserved in advanced plaques; contractile gene expression reduced in human lesions.	Skalli et al., J Cell Biol 1986 [24] Aikawa et al., Circ Res 1993 [27] Wirka et al., Nat Med 2019 [18]	Canonical markers often decline in late-stage disease, making lineage assignment difficult.
Fibroblast-like/Fibromyocyte	*FN1*, *COL1A1*, *LUM*, *DCN*, *BGN*, OPG *(TNFRSF11B)*	Robustly detected in murine lineage tracing; proliferative ECM-producing cells forming the fibrous cap.	Detected in human scRNA-seq datasets and spatial transcriptomics; linked to cap stability.	Wirka et al., Nat Med 2019 [18] Pan et al., Circulation 2020 [42] Alencar et al., Circulation 2020 [47]	Highly conserved state across species; dominant in stable lesions.
Myofibroblast-like	*COL1A1*, *COL3A1*, *MMP2*, *MMP9*, *OPN (SPP1)*	Observed in murine injury and atherosclerosis models; derived from dedifferentiated SMCs.	Confirmed in human coronary plaque datasets; shares ECM-remodelling and proliferative features.	Shankman et al., Nat Med 2015 [10] Pan et al., Circulation 2020 [42]	Boundaries between “synthetic” and “fibroblast-like” phenotypes are context-dependent.
Mesenchymal-like	KLF4-linked modules; ECM and cytoskeletal genes	Identified in *Myh11*-CreERT2; *ApoE*^−^/^−^ lineage tracing as intermediate transition state.	Confirmed in human coronary scRNA-seq datasets showing partial contractile and mesenchymal programmes.	Wirka et al., Nat Med 2019 [18] Alencar et al., Circulation 2020 [47]	May represent a transient state rather than a stable end-state.
Macrophage-like/Foam-cell-like	*LGALS3*, *CD68*, *ABCA1*, *LPL*	Observed in *ApoE*^−^/^−^ mouse lesions derived from SMCs; fate-mapping confirms SMC origin.	Identified in human plaques but rare; some clusters phenotypically resemble macrophages while retaining SMC markers.	Allahverdian et al., Circulation 2014 [54] Wang et al., Arterioscler Thromb Vasc Biol 2019 [43] Li et al., Cell Discovery 2021 [53]	Frequency in humans remains debated due to lineage ambiguity and marker overlap with myeloid cells.
Osteogenic-like	*RUNX2*, *OPN (SPP1)*, *BGLAP*, *ALPL*	Strong evidence from murine lineage tracing; enriched in advanced calcified lesions.	Detected in human coronary arteries and calcified plaques; associated with plaque instability.	Speer et al., Circ Res 2009 [56] Sun et al., Circ Res 2012 [57] Alsaigh et al., Comm Bio 2022 [55]	Indicates shared mechanism of SMC-driven calcification across species.
Chondrogenic-like	*SOX9*, *ACAN*, *COL2A1*	Reported in murine lineage tracing during atherosclerosis progression.	Present in human single-cell datasets; contributes to proteoglycan-rich ECM and plaque stiffness.	Speer et al., Circ Res 2009 [56] Xu et al., Biochem Biophys Res Commun 2012 [58] Pan et al., Circulation 2020 [42]	Overlaps transcriptionally with osteogenic programmes; context-dependent activation.
Adipocyte-like	*PLIN2*, *FABP4*, *ADIPOQ* (low)	Detected in mice (Sca1^+^ SMCs, though lineage origin debated).	No clear human homolog (Sca1 is murine-specific); relevance uncertain.	Pan et al., Circulation 2020 [42] Mosquera et al., Cell Reports 2023 [50]	May represent metabolic reprogramming rather than true adipogenic differentiation.
Intermediate/Transitional (SEM)	Mixed *ACTA2^+^/LGALS3*^+^ modules	Observed in high-resolution murine lineage tracing between contractile and fibroblast-like states.	Human scRNA-seq datasets suggest similar intermediate phenotypes, though less clearly resolved.	Wirka et al., Nat Med 2019 [18] Pan et al., Circulation 2020 [42]	Likely under-detected in human studies due to cross-sectional sampling.
EndMT-adjacent/Hybrid EC–SMC	*KDR*, *PECAM1*, *ACTA2* (co-expression)	Occasionally noted in murine atherosclerosis at lesion borders.	Human carotid single-cell atlases reveal EC clusters co-expressing SMC markers.	Depuydt et al., Circ Res 2020 [48] Pan et al., Circulation 2020 [42]	Represents a boundary phenotype rather than a canonical SMC fate.

## 3. Regulatory Mechanisms and Paradigm Shift in SMC Plasticity

Having outlined the emerging diversity of SMC phenotypes, we move on to summarise the current known regulatory mechanisms that underlie phenotypic plasticity, a key consideration for modelling in vitro.

### 3.1. Transcriptional Control of SMC Identity

#### 3.1.1. Maintenance of the Contractile State

The transcription factor, serum response factor (*SRF*), is a well-established regulator of SMC phenotypic plasticity, known to regulate the expression of SMC contractile genes such as *ACTA2*, *CNN1*, *MYH11* and *TAGLN* [59]. Most of these canonical marker genes contain two or more CArG (SRF motif CC(A/T)_6GG) elements within their promoters and/or intronic enhancers, which are bound by SRF to drive their SMC-enriched expression [60]. SRF is vital for smooth muscle development, differentiation and homeostasis [61,62,63]. SRF depletion has been shown to reduce the level of SMC marker expression, and to regulate proliferation and migration in a context-dependent manner [64,65]. Whilst SRF is an established regulator of the SMC contractile markers, due to its lack of specificity for SMCs, it cannot by itself be the master regulator of the contractile phenotype but depends on muscle-specific coactivators, such as myocardin (MYOCD) and myocardin-related transcription factors (MRTFs) [66].

MYOCD, a muscle-restricted co-activator of SRF with cardiac- and SMC-biased activity, has been implicated as an essential gene in cardiovascular muscle development, besides the regulation and maintenance of the contractile phenotype of SMCs [67,68]. MYOCD’s role as a master regulator of SMC differentiation is supported by gain-of-function experiments demonstrating that MYOCD is sufficient to confer structural and functional SMC characteristics and promote a mature contractile SMC phenotype in vitro [69,70]. It also protects SMCs from vascular inflammation and inhibits the transition to a pro-inflammatory phenotype at the onset of vascular disease [71]. Conversely, depletion of *MYOCD* expression results in consistent findings of a rapid decrease in contractile marker expression and upregulation of ECM proteins, demonstrating the necessity of MYOCD for vascular homeostasis and maintenance of the contractile phenotype in vivo [72,73]. Whilst SRF expression is essential for CArG element recognition in a non-SMC-specific manner, MYOCD confers the selectivity for SMCs, hence providing robust transcriptional regulation of the contractile programme [74,75].

Further regulators of the SRF-MYOCD axis are the co-activators, MRTF-A and MRTF-B (also referred to as MKL1/MKL2), which serve as SRF-dependent regulators of contractile genes. In contrast to MYOCD, MRTFs are expressed in a wide variety of tissues, both during embryonic development and in adulthood [76]. Their regulatory action is achieved through coupling actin dynamics to SRF-mediated CArG-dependent transcription [77,78], providing a mechanosensitive link with mechanical cues, such as ECM composition and stiffness, through the regulation of cytoskeletal arrangement by upregulating the expression of contractile genes [79]. MRTF-B is essential for cardiovascular morphogenesis, vascular development, and differentiation of SMCs in vivo [80,81]. By contrast, MRTF-A promotes pathological vascular remodelling, driving neointima formation and the SMC phenotypic switch, with enhanced proliferation in a reactive oxygen species (ROS)–dependent manner [82,83].

Additional transcriptional regulators have been implicated in supporting the contractile programme of SMCs. Myocyte Enhancer Factor 2 (MEF2) also co-regulates the SRF-MYOCD module, including direct upregulation of *MYOCD*, in response to calcium signalling to sustain the expression of contractile markers [84,85]. YAP/TAZ–TEAD signalling modulates the MYOCD–SRF–CArG complex in a context-dependent manner, coupling mechanical cues to cytoskeletal gene programmes to exert a mechanoprotective role [86,87]. Notch signalling is also essential for the differentiation and maturation of vascular SMCs, particularly Notch3, acting via RBPJ/HEY signalling, which supports the contractile programme in development and disease [88,89,90]. Furthermore, GATA6 has been found to support the maintenance of the differentiated contractile SMC phenotype [91,92]. While these axes are less universally conserved than SRF–MYOCD–MRTF, together they stabilise contractile identity and tune its sensitivity to environmental inputs.

Together, the SRF–MYOCD–MRTF core, modulated by MEF2, NOTCH3, GATA6 and mechanosensitive YAP/TAZ, stabilises the contractile programme. In disease, this module is progressively dampened by genetic and epigenetic regulation, alongside environmental cues and disease-related stimuli, priming SMCs for phenotypic modulation.

#### 3.1.2. Transcriptional Regulation Driving Phenotypic Plasticity

In phenotypic plasticity, the SRF-MYOCD-MRTF module is attenuated by various disease-induced signals, such as PDGF-BB, oxidised LDL (oxLDL), and inflammatory cytokines. Upregulation of plasticity-associated transcription factors then drives transitions into distinct synthetic states (Figure 2; Table 2), with a broader spectrum observed in mice and a more selective representation in human disease.

One of the best characterised transcriptional drivers of the synthetic state in atherosclerotic plaque genesis is Krüppel-like factor 4 (KLF4). *KLF4* expression is upregulated by several disease-related factors, such as PDGF-BB, oxidised phospholipids, and inflammatory cytokines, such as TNFα [93,94,95]. KLF4 represses the SRF-MYOCD contractile axis, downregulating the expression of contractile marker genes, *ACTA2*, *TAGLN* and *MYH11* [93]. Furthermore, recurrently upregulated KLF4 effectors include inflammatory markers *LGALS3* and *CD68*, the inflammatory ECM cytokine OPN, and matrix remodelling regulators, *MMP9* and *MMP3*, as well as *FN1* and *COL1A1* [10,42,47]. In vivo, SMC-specific deletion of *Klf4* reduces plaque size, the number of macrophages and increases fibrous cap thickness and stability [10]. Beyond an established role in cytokine signalling, KLF4 has been linked to the regulation of a macrophage-like state through excess cholesterol-induced endoplasmic reticulum stress in animal models [52]. Single-cell RNA-seq analysis revealed KLF4 dependency for an osteogenic phenotype, likely contributing to plaque calcification and destabilisation [47]. In integrative human single-cell analyses, KLF4-associated signatures are primarily enriched in mesenchymal-like SMC states, suggesting it is not a uniform driver of the phenotypic switch across all SMC phenotypes [32,50].

Another key transcriptional regulator of SMC plasticity, TCF21, emerged from a large-scale meta-analysis of GWAS that sought to define a CAD locus [96]. Lineage tracing studies have shown that *Tcf21*-expressing cells give rise to fibroblasts and SMCs during coronary artery development, consistent with its role in regulating mesenchymal differentiation. In the adult vessel wall, *Tcf21*-expressing cells contribute to the formation of the fibrous cap, potentially stabilising lesions [97]. Mechanistic insight emerged from human CASMC studies, which demonstrated that TCF21 blocks the association of the MYOCD-SRF complex, downregulating the contractile programme and canonical markers [19]. Further lineage tracing studies, supported by scRNA-seq from diseased human coronary arteries, identified TCF21 as a driver of a unique fibroblast-like phenotype, termed fibromyocytes. Loss of *Tcf21* was shown in these studies to reduce fibrous cap size and hence compromise plaque stability [18]. These findings highlight TCF21 as a regulator of SMC identity, providing a link between CAD risk variants and SMC plasticity across species [49]. Whilst TCF21 may promote fibrous cap formation, the fibromyocyte state reflects a deviation from the contractile programme and loss of SMC identity; hence, describing TCF21 as protective oversimplifies its role.

ELK1, an ETS domain transcription factor, is activated in response to PDGF-BB via ERK/MAPK signalling [98,99]. ELK1 has also been shown to disrupt the SRF-MYOCD axis by competing with MYOCD for the binding of CArG elements in SRF promoters, leading to the repression of the contractile gene programme of SMCs [100,101]. Recent findings from wire-injury animal models have demonstrated a link between enhanced ELK1 activity and dedifferentiation of SMCs to a highly proliferative synthetic phenotype [102].

OCT4, also known as POU5F1, the stem cell pluripotency factor, has additionally been implicated in modulating SMC phenotypic plasticity. *OCT4* expression was demonstrated to be reactivated in vascular SMCs in atherosclerosis. SMC conditional knockout studies have shown a reduction in fibrous cap size and an increase in necrotic core size, suggesting an atheroprotective role [103]. Supporting this, in acute vascular injury, *OCT4* was rapidly upregulated in SMCs, and its conditional deletion accelerated neointima formation, indicating a protective role in limiting proliferation [104]. By contrast, it was reported that enhanced neointimal growth after carotid artery injury occurs due to MMP2 activation, which stimulates SMC proliferation and migration [105]. This context dependence is further supported by isoform analysis, where both *OCT4A* and *OCT4B* were found to be upregulated in vascular SMCs during remodelling of pulmonary arteries, indicating multiple variants may contribute to SMC plasticity [106]. Integrative human transcriptomic analyses further support this context dependence, showing OCT4-associated programmes enriched in fibroblast-like, matrix-regulating states, in contrast to KLF4′s inflammatory and osteogenic biases [47]. Overall, OCT4 appears to have a context-dependent role in re-routing SMCs toward fibrotic- and matrix-associated states [107].

RUNX2 and SOX9 are characterised as regulators of the osteogenic and chondrogenic phenotypes, influencing calcification of the vessel wall. Lineage tracing studies showed that SMCs give rise to an osteochondrogenic precursor population that upregulates *Runx2* and downregulates *Myocd* expression [56]. SMC-specific deletion of *Runx2* in animal models resulted in the suppression of vascular calcification, establishing *Runx2* as a requisite for the osteogenic phenotypic switch [57]. *Sox9* overexpression in SMCs was shown to suppress the *Myocd*-regulated expression of contractile genes and promote transdifferentiation into chondrocytes [58]. Recent findings indicate that *Sox9* contributes to ECM stiffening and vascular ageing, thereby indirectly promoting calcification in vivo [108]. Single-cell and integrative human data analysis confirms the presence of RUNX2-associated osteogenic and SOX9-associated chondrogenic SMC-derived clusters in atherosclerotic plaques [32,50].

Together, these transcription factors redirect SMC fate from the SRF-MYOCD-MRTF contractile programme towards a broad spectrum of alternative SMC phenotypes. It is essential to reiterate that, while TCF21- and KLF4-linked fibroblast-like programmes appear conserved in human plaques, the regulation of macrophage-like and intermediate phenotypes remains more robustly defined in murine models. Beyond transcriptional control, chromatin remodelling and noncoding RNAs also influence SMC fate, but a detailed discussion is outside the scope of this review.

### 3.2. Extracellular Cues and Disease-Relevant Stimuli

#### 3.2.1. ECM Composition

ECM components have been implicated in SMC phenotypic switching, as they provide critical environmental cues; however, their precise roles are not yet fully resolved.

Fibronectin (FN), an ECM glycoprotein that binds cell surface receptors, is one of the most thoroughly studied coating matrices employed in vitro. Early work with plasma FN coating has associated it with increased synthetic SMC modulation, resulting in enhanced cell migration and proliferation [109,110,111]. Mechanistic insight from cultured arterial SMCs demonstrated that this occurs via ERK signalling [112]. However, FN has also been shown to stabilise cell adhesion and encourage survival, highlighting a context-dependent role [109,113].

Laminin, a common basement membrane protein in the healthy vessel wall, has been shown to support the maintenance of the contractile phenotype, as evidenced by the preservation of elongated morphology and contractile marker expression [109,114]. Therefore, laminin is considered a stabilising factor in SMC differentiation.

Conversely, elastin, another key component of arterial walls, is downregulated during phenotypic switching, contributing, along with the upregulation of collagen, to arterial stiffening [115,116].

Collagens have a more diverse role in regulating phenotypic modulation. Type IV collagen is normally enriched in the basement membrane and associated with contractility, whilst type I and III collagens are often found to be upregulated in atherosclerosis [117,118,119]. The roles of collagen isoforms in vitro were demonstrated by Orr et al., who found that type I collagen promoted proliferation, while type IV contributed to maintaining the contractile phenotype [120]. Gelatine, a denatured form of collagen, has been widely used as a coating matrix in vitro due to ease of handling and low cost. While it supports adhesion and growth, it lacks defined instructive signalling and in vivo relevance to recapitulate the SMC environment [121].

Beyond the composition of the ECM, its stiffness is a key driver of SMC phenotypic plasticity. In atherosclerosis, the stiffening of the vessel wall is driven by collagen deposition and the fragmentation of elastin [116]. This stiff environment was recapitulated in vitro by plating SMCs on substrates of different stiffness created by collagen and FN, which showed effects on cytoskeletal arrangement and migration [122].

In the human context, medial SMCs are embedded in a type IV collagen, laminin-411/511, perlecan and nidogen-rich basement membrane, supported by the adjacent elastin [117,118]. Alterations to ECM composition, such as increased fibronectin or collagen types I and III, are linked to the pathological remodelling of the vessel wall [119].

#### 3.2.2. Soluble Factors

Alongside transcriptional regulation, soluble factors in the diseased vessel wall act as key extrinsic drivers of SMC phenotypic switching.

In response to endothelial injury and inflammation, PDGF-BB expression is upregulated and secreted by activated immune cells, platelets, and the endothelium, promoting the dedifferentiation, cytoskeletal remodelling, proliferation, and migration of SMCs [123,124,125]. The hyper-stimulated SMCs migrate to the intima and contribute to intimal thickening during the initiation of atherosclerotic lesion formation [30,126]. SMCs are further sensitised to PDGF-BB during atherosclerosis through the upregulation of α- and β-receptors and attenuated lysosomal degradation [127]. Activation of PDGFR-β induces signalling via the ERK/MAPK, PI3K/AKT, JAK/STAT, Notch/MMP and JNK cascades [128,129,130].

Beyond the canonical PDGF-BB effects, its enhanced secretion leads to a coordinated regulation of the SMC phenotypic switch, characterised by the downregulation of contractile markers, upregulation of ECM components and matrix remodelling enzymes, such as MMPs [30,94,98]. In vivo, this sustained secretion of PDGF-BB drives neointimal formation in response to vascular injury and prompts the clonal expansion of SMCs within atherosclerotic plaques [46,107]. These in vitro and in vivo findings establish PDGF-BB as a prominent stimulus underlying SMC phenotypic plasticity across human and murine models of atherosclerotic disease, in part by converging on the transcriptional regulators described above.

In contrast to the clear one-sided role of PDGF signalling in SMC plasticity, TGF-β signalling has highly context-dependent roles. In vascular homeostasis, TGF-β–SMAD2/3 signalling supports the contractile phenotype by reinforcing the SRF–MYOCD–driven contractile gene expression [98]. However, following vascular injury, dysregulated TGF-β signalling drives ECM remodelling, enhancing collagen and proteoglycan deposition, leading to intimal thickening and stiffening of the vessel wall [116]. These changes in the mechanical environment promote the downregulation of the contractile programme and upregulate a pro-fibrotic gene expression in SMCs [79]. In atherosclerosis, this pro-fibrotic reprogramming can stabilise the fibrous cap by increasing the number of ECM-producing fibroblast-like cells [18]. However, excessive TGF-β also enhances pathological fibrosis and disrupts vascular homeostasis, thereby exacerbating the progression of atherosclerotic disease [79,126,131]. Taken together, TGF-β is a crucial regulator of SMC phenotypic plasticity, with a highly context-dependent role in response to the surrounding environmental cues.

Other commonly present soluble factors in atherosclerotic conditions that impact SMC fate include oxLDL and modified cholesterol. Uptake of these molecules triggers ROS accumulation, leading to oxidative and endoplasmic reticulum stress in SMCs [94]. This leads to downregulation of the SRF-MYOCD contractile programme, while inducing the expression of inflammatory and matrix remodelling gene signatures [52]. In murine lineage tracing studies, cholesterol uptake by SMCs is documented to drive the transition of SMCs into foam cell- and macrophage-like cells [43,53,54]. However, human studies indicate a more restricted contribution of SMC-derived foam cells [54]. Furthermore, excess cholesterol uptake and oxLDL exposure can drive osteogenic conversion of SMCs and contribute to calcification of the vessel wall [47,55]. Additional inflammatory mediators contributing to SMC plasticity in atherosclerosis are driven by cytokine signalling, especially IL-1β and TNF-α. Both cytokines have a dual impact on SMCs, repressing the contractile programme, while upregulating the secretion of inflammatory cytokines, and MMPs, leading to enhanced matrix degradation and migration, and ultimately, atherosclerotic lesion instability and expansion [14,94,95,132].

Collectively, these soluble mediators—PDGF-BB, TGF-β, oxLDL, IL-1β, and TNF-α —are key regulators of SMC phenotypic plasticity and atherosclerotic disease progression. They are not only central modulators in experimental settings, but they have also been implicated as therapeutic targets in human CVDs [107,133].

#### 3.2.3. Crosstalk with Neighbouring Cells

The involvement of SMCs in control of vascular function, by stabilising the vessel wall and regulating vascular tone, is shaped by continuous interactions with neighbouring ECs and immune cells. Crosstalk between cell types is essential for maintaining homeostasis, but it can amplify pathogenic signalling in the context of vascular injury.

Signalling by ECs through both direct contact and paracrine routes is central for the regulation of SMC behaviour. In homeostatic conditions, ECs provide cues to maintain the SMC contractile identity, a controlled rate of proliferation and migration, and a state of SMC quiescence [134,135]. However, during endothelial dysfunction caused by inflammation or disrupted laminar flow, crosstalk between SMCs and ECs is disrupted. Signalling from dysfunctional ECs contributes to the dedifferentiation of SMCs, altering their behaviour by enhancing migration and proliferation, thereby contributing to neointima formation [136,137,138,139]. Recent in vitro models that preserve the EC-SMC cross-communication have shown that signalling across cell types dynamically regulates gene expression patterns and ECM remodelling, highlighting the importance of this axis in both health and disease [140,141].

Other neighbouring cell types that regulate SMCs behaviour include immune cells, particularly macrophages, T cells and platelets. These cells exert strong modulatory effects on the SMC phenotype, primarily through the secretion of inflammatory cytokines and growth factors, such as those described above. Signalling from these cells represses the contractile programme whilst inducing an inflammatory phenotypic state in SMCs [142,143]. In turn, the modulated SMCs contribute to the inflammation within atherosclerotic plaques by secreting chemokines and cytokines, as well as ECM fragments that lead to further recruitment and retention of immune cells in the lesion [144]. Ultimately, cross-communication between SMCs and immune cells amplifies the inflammatory environment, enhancing the size of the necrotic core, leading to lesion destabilisation and progression.

Together, these intercellular interactions highlight the complexity of SMC fate regulation. While cooperative crosstalk sustains vascular stability in homeostasis, in disease conditions, it can drive pathological remodelling of the vessel wall.

### 3.3. Implications for Modelling

Regulation of SMC fate spans layers of control, from transcriptional networks, ECM composition, soluble mediators, to intercellular crosstalk. These signals act jointly to maintain or dedifferentiate the contractile SMC phenotype. Recapitulating the complexity of SMC fate regulation remains a major challenge. In vitro systems are limited by the reductionist nature, where often, only a single stimulus is investigated in an artificial context. In vivo models may better capture the structural complexities and systemic influence, but they are largely limited by the inability to isolate individual mechanisms for investigation, and by the species gap in translation [145,146]. Murine models often fail to recapitulate the trajectories observed in human plaques, highlighting fundamental differences in the prevalence and nature of SMC-derived cell states across species. These divergences underscore the necessity of designing and validating models that are grounded in human-relevant biology [18,48,50].

**Table 2 cells-14-01913-t002:** Summary of key regulators and associated SMC phenotypes. Regulators of smooth muscle cell (SMC) phenotypic plasticity and their primary mode of action, encompassing transcriptional control, soluble mediators, extracellular matrix (ECM) cues, and intercellular crosstalk. The SRF–MYOCD–MRTF core complex and its modulators (MEF2, YAP/TAZ, NOTCH3, GATA6) sustain the contractile programme under homeostatic conditions, whereas transcriptional repressors and inducers of alternative gene modules (KLF4, TCF21, ELK1, OCT4, RUNX2, SOX9) mediate distinct disease-associated transitions. Environmental and paracrine stimuli, including PDGF–PDGFRβ, TGF-β, oxLDL, and inflammatory cytokines, further modulate these transcriptional networks, promoting fibroblast-like, macrophage-like, or osteogenic reprogramming. ECM composition and stiffness, together with signalling from endothelial and immune cells, integrate mechanical and inflammatory inputs that collectively shape SMC fate. Table entries summarise each regulator’s mechanism, the predominant associated phenotype(s), and key supporting evidence from animal and human studies.

Regulator/Pathway	Primary Function/Mechanism	Associated SMC Phenotype(s)	Key Evidence
SRF–MYOCD –MRTF axis	Core transcriptional module driving contractile gene expression (*ACTA2*, *TAGLN*, *CNN1*, *MYH11*) via CArG-box regulation. MYOCD confers SMC specificity; MRTFs link actin dynamics to transcription.	Contractile (baseline); loss promotes synthetic transition.	SRF/MYOCD depletion reduces contractile markers and increases ECM genes in vivo and in vitro [59,60,61,62,63,64,65,66,67,68,69,70,71,72,73,74,75,76,77,78,79].
MEF2	Co-activator of the SRF–MYOCD module; integrates Ca^2+^ signalling to sustain the contractile programme.	Contractile maintenance.	Upregulates *MYOCD* and contractile genes in Ca^2+^-dependent manner [84,85].
YAP/TAZ (TEAD-dependent)	Mechanosensitive regulators coupling cytoskeletal tension and ECM stiffness to SRF–MYOCD signalling.	Contractile maintenance under mechanical stress.	Activation preserves cytoskeletal integrity; dysregulation favours synthetic transition [86,87].
NOTCH3 (RBPJ/HEY)	Promotes differentiation and maturation of vascular SMCs; supports contractile identity.	Contractile maintenance.	Genetic loss impairs vascular maturation and induces synthetic features in mice [88,89,90].
GATA6	Transcription factor sustaining differentiated contractile state.	Contractile.	Reduced expression linked to dedifferentiation in vascular disease models [91,92].
KLF4	Represses SRF–MYOCD axis; induces inflammatory and matrix genes (*FN1*, *COL1A1*, *MMPs*, *OPN*). Activated by PDGF-BB, oxLDL, TNF-α.	Macrophage-like, osteogenic, mesenchymal-like.	SMC-specific *Klf4* deletion reduces lesion size and enhances fibrous-cap stability; human scRNA-seq links to mesenchymal signatures [93,94,95].
TCF21	Inhibits MYOCD–SRF interaction to suppress the contractile programme; drives fibroblast-like transition.	Fibroblast-like (fibromyocyte).	Lineage tracing and human scRNA-seq show cap-forming fibromyocytes dependent on *TCF21* [18,19,96,97].
ELK1	PDGF-BB-induced ETS factor competing with MYOCD for SRF binding, repressing contractile genes.	Synthetic (proliferative).	Enhanced ELK1 activity drives SMC dedifferentiation and neointimal growth in mice [98,99,100,101,102].
OCT4 (POU5F1)	Reactivated pluripotency factor modulating matrix and fibrosis programmes; isoform-specific actions.	Fibroblast-like/matrix-associated states.	Conditional knockout reduces fibrous cap size; human data show enrichment in matrix-regulating clusters; context-dependent [103,104,105,106].
RUNX2	Master regulator of osteogenic conversion and calcification; suppresses MYOCD.	Osteogenic-like.	SMC-specific *Runx2* deletion reduces vascular calcification; osteogenic clusters in human plaques [56,57].
SOX9	Promotes chondrogenic transition and ECM stiffening; antagonises MYOCD.	Chondrogenic-like/fibrotic.	Overexpression induces chondrogenic markers and stiff ECM in vivo and human arteries [58,108].
PDGF-BB–PDGFRβ	Canonical inducer of SMC phenotypic switch via ERK/MAPK, PI3K/AKT, JAK/STAT, Notch/MMP, JNK pathways; stimulates migration and proliferation.	Synthetic/fibroblast-like.	In vitro and in vivo evidence for SMC dedifferentiation and clonal expansion in lesions [123,124,125,128,129,130].
TGF-β–SMAD2/3	Context-dependent: supports contractile programme in homeostasis, drives fibrosis after injury.	Contractile (maintenance) or Fibroblast-like (pro-fibrotic).	Dual role supported by animal and human data linking to fibrous-cap stability [79,126,131].
oxLDL/Cholesterol uptake	Induces oxidative and ER stress; activates inflammatory and matrix genes.	Macrophage-like and osteogenic-like.	Murine lineage tracing shows SMC-derived foam cells; human data support restricted conversion [52,94].
IL-1β/TNF-α	Cytokine signalling that represses contractile markers and activates MMPs and inflammatory genes.	Inflammatory/macrophage-like.	Demonstrated in SMC cultures and atherosclerotic murine models [14,94,95,132].
ECM composition (Fibronectin, Laminin, Collagen, Elastin)	ECM proteins govern adhesion, migration and phenotype; fibronectin and collagen I/III promote synthetic state; laminin and collagen IV preserve contractility.	Contractile or Synthetic (dependent on matrix type).	In vitro matrix-coating and in vivo remodelling studies show context-dependent effects [109,110,111,112,113,114,115,116,117,118,119].
ECM stiffness/mechanotransduction	Increased stiffness drives cytoskeletal reorganisation and synthetic switch.	Synthetic/mesenchymal-like.	Substrate-stiffness models and atherosclerotic tissue data support mechanosensitive regulation [116,122].
Endothelial crosstalk	EC-derived signals maintain SMC quiescence and contractility; dysfunction induces dedifferentiation and migration.	Contractile (homeostasis) or Synthetic (pathology).	Co-culture and injury models demonstrate bidirectional influence on SMC fate [134,135,136,137,138,139].
Immune cell crosstalk	Macrophage and T-cell cytokines (IL-1β, TNF-α) promote inflammatory and migratory SMC states.	Inflammatory/macrophage-like.	Murine and human plaque data show reciprocal inflammatory amplification [142,143].

## 4. In Vitro Models of SMC Plasticity: From Monocultures to 3D Systems

Building on the regulatory layers outlined above, experimental models are crucial for interrogating SMC plasticity under defined conditions. The challenge lies in balancing reductionist approaches that provide mechanistic clarity with more complex multicellular systems that better recapitulate the vascular environment. In this section, we evaluate the spectrum of models available and assess their capacity to capture human-relevant features of SMC fate regulation.

### 4.1. What Makes a Good Model?

Before assessing individual systems, it is essential to establish what would be considered a robust, reliable, and relevant model of SMC plasticity. Many studies rely on a single readout to evidence SMC phenotypic switching, such as the downregulation of a contractile marker at the mRNA or protein level, or focus solely on a functional readout, like proliferation or migration [10,14,107]. Whilst each measure can provide important snapshots into the biology, there is a large risk of oversimplifying the complex regulatory layers. To deem a model useful, plasticity should ideally be assessed across multiple complementary layers spanning transcriptional, translational, and functional levels, linking molecular changes to alterations in cell behaviour [54,147].

Another crucial, often overlooked, step is ensuring that the cells display a robust contractile phenotype prior to simulation of disease. Ensuring the baseline condition is a contractile SMC guarantees the findings do not only reflect adaptation to culture conditions. Therefore, a good model requires rigorous initial characterisation, ideally benchmarking against human datasets to reduce artefacts and maximise relevance.

### 4.2. Simple Monocultures

Primary SMC monocultures remain the most widely utilised model for investigating phenotypic plasticity. Their main appeal lies in the precise control over experimental conditions, allowing for the examination of individual factors, stimuli, or mechanical cues to study their effects in isolation. Pioneering work on SMC plasticity induced by PDGF-BB in monoculture has been instrumental in understanding the contractile to synthetic switch [12,126]. They are indispensable as a tool for mechanistic studies, providing a scalable, amenable, prompt platform for studies utilising genetic perturbations, such as siRNA or CRISPR-Cas9 systems, and pharmacological screening, especially in high-throughput formats, where more complex models would be impractical [131].

A well-recognised limitation of monocultures is that primary SMCs rapidly dedifferentiate upon isolation; therefore, cultures invariably begin from a partially synthetic state, highlighted by the proliferative nature of SMCs in culture. To preserve their native phenotype as long as possible, optimisation of culture conditions, including culture media supplementation, FBS concentration, heparin supplementation, cell density and ECM coating matrix, can be carried out to lengthen the lifespan of contractile SMCs in culture [28,148,149]. Another key variable is sourcing the primary human cells. Their availability is often limited, variability across donors can distort findings, and, nonetheless, they are expensive. In contrast, rodent primary SMCs are more accessible, and reproducibility is less of an issue; however, due to the differences in lineage origins and gene regulatory pathways, their translational relevance is questionable [150]. In practice, rodent primary SMCs are often easier to expand and maintain than human primary SMCs, which frequently show slower initial growth and greater donor-to-donor variability. These practical differences in culture behaviour can influence experimental design and interpretation. As an accessible alternative, immortalised SMC lines exist. The most widely used are the rodent A7r5 and MOVAS lines, while human lines such as T/G HA-VSMC are also available [151,152,153]. Whilst they overcome the issues of availability, cost, reproducibility, and limited doubling times, due to their immortalisation, their phenotypes largely deviate from those of primary cells, particularly in cytoskeletal organisation and contractile marker expression, which limits their disease relevance. Additional challenges arise from the lack of reliable surface markers to distinguish contractile SMCs, which limits sorting and purification strategies. As a result, eliminating fibroblast contamination from primary isolates is virtually impossible and remains a confounding factor that influences transcriptional and functional readouts.

Overall, human primary SMC monocultures remain valuable and justifiable for reductionist mechanistic studies. However, their limitations must be acknowledged, and findings ideally validated against human tissue data or sequencing datasets to ensure disease relevance.

### 4.3. Human iPSC-Derived SMCs

Induced pluripotent stem cell (iPSC)-derived SMCs are an emerging tool to study SMC plasticity in a human-relevant context. Their popularity stems from being a renewable, scalable, and individual patient-specific source for genetic and mechanistic studies of SMC biology. However, being more representative of a foetal-like state at baseline, their ability to recapitulate a mature contractile SMC phenotype is questionable [154,155].

Despite significant improvements in differentiation protocols for creating SMCs from human iPSCs, differentiating robust, pure SMC cultures with mature, functional phenotypes remains challenging. Protocols define differentiation through different developmental origins, including via the epicardium, neural crest, and paraxial/splanchnic mesoderm, to mimic in vivo trajectories and derive organotypic coronary, aortic arch and descending aortic SMCs, respectively [156,157,158]. To address the maturity and purity concerns associated with incomplete cytoskeletal organisation and limited expression of contractile proteins, several strategies have been employed. For example, *ACTA2*- and *MYH11*-reporter iPSC lines have been created to select for SMCs that have an activated contractile programme [159,160]. Optimisation of culture conditions, such as through TGF-β1, retinoic acid supplementation, and serum starvation, has also been shown to increase the efficiency of mature SMC differentiation and reduce the presence of synthetic SMCs [161,162,163]. To further ensure a true contractile SMC phenotype, maturity is assessed beyond the transcriptional and translational levels by functional assay for contractility, calcium flux and electrophysiological responsiveness [163,164,165]. Considering how immaturity complicates the interrogation and interpretation of SMC plasticity, for iPSC-derived SMC to reflect a true phenotypic transition rather than a developmental maturation, such steps must be taken. Establishing a contractile baseline phenotype with high purity is a prerequisite for iPSC-derived SMC studies. Subsequent steps to ensure the model reflects a mature in vivo state should involve benchmarking in vitro states against human single-cell and tissue data, which will be covered in Section 5. Under defined cues, iPSC-SMCs up- or down-regulate contractile programmes. PDGF-BB drives a phenotypic transition towards a synthetic state with down-regulation of contractile markers in human iPSC-SMCs, establishing these cultures as a workable platform to interrogate switching mechanisms [163,164].

One of the greatest strengths of iPSC-derived SMC models is the powerful ability to investigate the interplay between genetic variations and phenotypic modulation. For instance, patient-derived iPSC-SMCs have been shown to capture the genetic determinants of plasticity, such as dysregulated ECM remodelling and TGF-β signalling abnormalities in Marfan syndrome, or cytoskeletal defects and hyperproliferative and migratory rates of SMCs in Supravalvular Aortic Stenosis Syndrome [166,167]. The translational potential of iPSC studies spans beyond mechanistic and genetic studies through their incorporation into more complex vascular models. For instance, in vascular organoids and microfluidic models, their interaction with ECs, fibroblasts and immune cells can better recapitulate multicellular niches, to improve platforms for investigation of SMC plasticity [146,168,169,170].

Taken together, human iPSC-derived SMCs are a powerful translational tool that enables genetic tractability, patient-specific investigation, and scalability of SMC plasticity studies, with the caveat that their immaturity and incomplete representation of adult states limit the interpretation of the data.

### 4.4. Co-Cultures

As described in Section 3.2.3, neighbouring cells—ECs, fibroblasts, immune cells—largely influence SMC phenotypic plasticity under in vivo conditions. To model a more physiologically relevant state, considerable efforts have been made to recapitulate complex multicellular environments in in vitro co-culture systems [142]. Below, we discuss and evaluate the co-culture designs based on their interaction topology, dynamics, and cellular composition, highlighting the benefits and disadvantages.

#### 4.4.1. EC-SMC Co-Cultures

There are several types of model, each recapitulating a different aspect of the intercellular cross- communication, to model and investigate their crosstalk is vital in the assessment of atherosclerotic plaque stability [171].

In direct contact models, EC-SMC crosstalk is facilitated by gap-junction signalling, through preserved myoendothelial junctions. These models aim to be representative of resistance vessels where EC–SMC communication occurs across internal elastic lamina fenestrations via myoendothelial junctions. Direct contact systems are beneficial in investigating Notch and Jagged signalling, where EC Jagged1 activates Notch3 in adjacent SMCs, stabilising the contractile phenotype [172,173]. Preserving contact between ECs and SMCs prolongs the maintenance of the contractile phenotype by reducing SMC proliferation and inflammatory activation [134,137].

To model larger arteries, where a basement membrane separates ECs from SMCs, cells can be seeded on the opposite sides of a porous membrane, such as in Transwell systems. Depending on the pore size, the formation of junctions between cell types can be allowed or limited for context-dependent investigations. Transwells are ideal for studying paracrine cues in isolation, such as PDGF-BB, TGF-β, or endothelin-1 (ET-1), as contact-dependent effects are excluded. Whilst Transwell systems move a step closer to recapitulating the intercellular communications, the width and composition of the membrane may limit physiological relevance. The variable pore size can be tuned to either restrict the passage to only soluble factors or allow migration of cells. Studies utilising Transwell systems have revealed how EC-derived cues influence SMC phenotype, inducing a transition from the contractile to synthetic state by paracrine signalling [171,174]. To create a more physiologically representative barrier, cells can be separated by laminin/collagen-IV–rich gels to emulate the intima–media interface, permitting diffusion of NO, ET-1 and PDGF while limiting gap-junction signalling [175]. Due to the key role of ECM composition in maintaining SMC phenotypes, the composition, stiffness, and architecture of the basement membrane are important for SMC identity.

Crosstalk between ECs and SMCs can be investigated without cellular contact, in a directional approach, by transferring conditioned media to study the effects of secretome signalling. Foundational work has shown that EC-conditioned media can induce DNA synthesis and proliferation, whilst reducing the expression of contractile marker proteins in SMCs [176,177]. This model can distinguish baseline EC secretome effects from injury-specific modulations in disease modelling.

Micropatterned 2D co-cultures, created by microcontact-printing stripes or stencil bilayers, enable precise control of EC–SMC contact length and alignment, allowing for the dissection of juxtacrine dose–responses [178]. These models are particularly valuable for studying cell alignment-dependent signalling and cell morphology [179].

Whilst the preservation of EC-SMC cross-communication is an important step towards in vivo-relevant modelling of the vasculature, it comes with several challenges and limitations. Media compatibility is a significant challenge, as ECs and SMCs require different media with various supplementary factors that can influence the fate of SMCs. Because SMCs vastly outnumber ECs in the vessel wall, co-cultures should report EC:SMC seeding ratios and verify lineage proportions at endpoint. With membrane separation, constraints due to composition, thickness, and pore sizes limit physiological fidelity. The source of human primary cells is generally a limiting factor in cultures, but in the case of co-cultures, it complicates the process further by the need to pair ECs and SMCs from matched donor vascular beds. Otherwise, confounding factors such as age, disease stage, and sex can have significant effects.

#### 4.4.2. SMC–Immune Co-Cultures

To model the inflammatory control of SMC plasticity, highly relevant for mimicking an atherosclerotic background, SMC–immune bi-culture models can be utilised. These formats allow investigation of leukocyte-derived cues that reprogram SMCs towards inflammatory and foam–cell–like phenotypes, as identified by lineage tracing and single-cell transcriptomics [43,51,52,53,54].

Direct contact models are crucial in studying the juxtacrine cues and synapse-like interactions between macrophages, T cells, and SMCs, recapitulating effects that exceed paracrine signalling. Studies utilising these models consistently report loss of the contractile programme, through downregulation of contractile proteins αSMA and SM22α, along with a gain of inflammatory signatures through the upregulation of Mac-2 (Galectin-3), MMPs, and cytokines [180,181]. With additional lipid exposure, findings from in vitro models align with the identification of foam-cell-like and inflammatory SMC states [182,183]. These models are vital in interrogating the contact-dependent regulation of matrix remodelling through MMP activity, lipid uptake and retention of SMCs, induction and stabilisation of the inflammatory state beyond cytokine activity [184]. In CD8+ T-cell and SMC co-cultures, direct contact drives SMC dedifferentiation to adopt features of macrophage-like and osteoblast-like phenotypes, shown by *Runx1*, *Spp1*, and *Bmp2* upregulation [185].

To isolate paracrine-only signalling effects driving divergent inflammatory versus synthetic responses, Transwell or insert-separated systems can be used to focus on responses to soluble mediators. Immune cells and SMCs can be plated on opposing sides of a well-defined porous membrane, where the material and thickness restrict the transfer of soluble factors and limit vesicle transport and migration. Studies employing this format for SMC-monocyte/macrophage models showed the paracrine induction of MMP-1/MMP-9, IL-6/TLR4, and even calcification under RANKL/M1-polarised cues [186,187,188,189].

One way immune-SMC systems that aim to study signalling without contact artefacts utilise directional conditioned-media swapping. In these models, macrophage-conditioned media were found to drive SMC proliferation and trigger activation of calcification programmes, and diabetic macrophage-conditioned media were reported to further increase proliferative rates [188,190].

SMC-immune cell bi-culture models are important for investigating their interactions and the regulation of SMC plasticity, but findings should be interpreted cautiously, given their limitations. By design, they omit the endothelial barrier, increasing the risk of in vitro artefacts; benchmarking to human plaque datasets is thus advisable [50]. Immune polarisation drift is commonly observed over 24 h of culturing, which can undermine the contrast between inflammatory and synthetic responses; hence, media refresh schedules, media-matched controls and starting and endpoint marker checks are essential [143]. Use of immortalised lines (e.g., THP-1) can misrepresent primary human T cells/macrophages; key findings should be validated in primary cells, ideally donor-matched [147]. Because SMCs dominate wall biomass in vivo, in vitro studies often adopt higher immune cell/SMC ratios to unmask signals, which should be reported explicitly and interpreted with caution. Finally, contact vs. paracrine mechanisms can be conflated in mixed cultures; pairing direct-contact with insert-separated and/or conditioned-media formats strengthens causal inference for soluble mediators vs. juxtacrine effects [186,187,188,189].

#### 4.4.3. EC–SMC–Immune Cell Tri-Cultures

Tri-culture models, featuring SMCs, ECs, and immune cells, are a step closer to recapitulating in vivo physiological relevance, particularly warranted for modelling endothelial damage-induced immune cell activation and the consequent SMC state changes that cannot be resolved in pairwise systems.

Practical and informative 2D implementation of the tri-cultures often use a tri-layer Transwell system, where ECs are grown on the apical side of the insert that can be activated with disease-relevant cues (such as TNFα, IL-1β, oxLDL) before introduction of immune cells, whilst SMCs are seeded at the basal side. This set-up allows investigation and quantification of adhesion, EC barrier integrity, transmigration across the EC layer, and SMC phenotypic state changes.

Recent findings from triple-cell studies showed that the inclusion of all three cell types drives a stronger SMC phenotypic shift, compared with mono/bi-cultures, characterised by great loss of contractile markers *ACTA2*, *CNN1*, and enhancement of inflammatory programmes demonstrated by upregulation of MMP1, MMP9, and various cytokines, and, where lipid is present, increased lipid uptake/foam index, highlighting the dependence of SMC plasticity on cues from both ECs and immune cells [143,144,145]. Interpretation of findings should be guided by SMC/EC/immune cell ratios, cell density reports, insert material, thickness and pore size definitions, exposure windows, culture media matching and refresh schedules, influencing immune polarisation.

### 4.5. 3D and Microfluidic Systems

Matrix architecture, stiffness, and permeability critically influence SMC state through mechanotransduction (SRF–MYOCD/MRTF; YAP/TAZ) and hypoxia or gradient effects, to activate ECs, which subsequently reprogram SMC phenotypes through EC-derived cues. Modelling that extends beyond 2D better recapitulates features of the fibrous cap, such as the ECM, calcifying niches, and permeability—and chemotaxis–dependent transmigration—which are difficult to capture in planar cultures.

#### 4.5.1. Spheroids and Aggregates

SMC spheroids develop a dense hypoxic core and a more contractile phenotype in the periphery, with a radial matrix and oxygen gradient. This setup enables comparison of core and rim SMC phenotypes, especially regarding hypoxia pathways. Incorporating both ECs and SMCs into self-assembling 3D structures enables investigation of crosstalk mechanisms regulating vascular remodelling and SMC plasticity [191]. For example, in human spheroid models incorporating SMCs, HIF-1α activation in the mural layer was found to be associated with osteogenic programme induction, primarily through RUNX2 activation, providing a 3D model to further examine calcification-relevant SMC states [192]. Spheroid models have been used to examine the contractile function of SMCs, demonstrating strong responses to ET-1. This platform provides a direct means to assess SMC plasticity, which is often challenging to study in planar cultures but is essential alongside phenotype markers [193]. Furthermore, their utility in drug screening pipelines has been demonstrated through quantitative characterisation of heterogeneous morphological responses to drug perturbations, shown by single-spheroid analysis [194]. Limitations to be considered in data interpretation include the core-rim heterogeneity and spheroid diameter, which can blur bulk readouts. Moreover, EC–SMC co-culture spheroids do not consistently self-organise into discrete endothelial and smooth-muscle layers; arrangements can be irregular or partially mixed, limiting structural fidelity to in vivo vessels [194]. Because ECM composition and stiffness are often undefined on non-adherent plastics, mechanotransduction inferences should be made judiciously unless matrix cues are controlled.

#### 4.5.2. Hydrogel-Embedded Constructs

Hydrogel-based models allow tailoring of the ECM composition and stiffness, key determinants of SMC plasticity, to recreate environments of a healthy vessel wall, or a more fibrotic plaque-like environment [195]. In many 2D and some 3D contexts, softer matrices favour contractile gene expression, whereas stiffer matrices promote synthetic programmes. In PEG-fibrinogen hydrogels, increasing matrix stiffness promoted the transition from contractile phenotype through enhanced RhoA activity, demonstrating mechanics-driven SMC phenotypic plasticity [196]. Conversely, in GelMA/alginate bioprinted constructs, a higher-solids blend with higher viscosity pre-gel and stiffer post-crosslink mechanics preserved viability, promoted spindle morphology and proliferation, and enhanced contractile marker expression vs. leaner gels, demonstrating that gel composition and stability gate SMC differentiation in 3D [197]. Therefore, findings should be interpreted in the context of polymer type, crosslinking, and stiffness, which profoundly influence SMC plasticity.

#### 4.5.3. Tissue-Engineered Vascular Constructs

Tissue-engineered rings and tubes aim to recapitulate the physiological organisation of cells around a lumen, often incorporating an EC inner layer, in a geometry that resembles the media and intima layers of the vessel. These models are particularly useful to investigate the effect of physiological pressure and blood flow with circumferentially aligned SMCs. Human iPSC-SMC engineered 3D tissue rings are a patient-specific, mechanistically robust and functional model that allows the investigation of proliferative diseases. They have been shown to recapitulate contractile SMC phenotypes that exhibit high contractility in response to agonists such as KCl [165]. These constructs also facilitate spatial analysis of SMC phenotype and matrix deposition within a vascular-like architecture, enabling the inclusion of additional vascular cell types such as ECs and fibroblasts, and supporting integration into organ-on-chip systems that more closely mirror the vessel wall [198].

#### 4.5.4. Microfluidic “Vessel-on-Chip”

Advanced 3D multicellular systems are progressively advancing towards recapitulating in vivo vascular structure and cellular cross-communication. Vessel-on-chip models enable real-time analysis of EC-SMC interactions and interrogation of the propagation of endothelial shear state responses that induce SMC phenotypic and matrix remodelling. Models utilising this set-up have shown that EC and SMC layers reproduce features of the arterial wall and demonstrate that EC activation promotes SMC migration and phenotypic modulation through paracrine signalling [170,199]. Incorporation of human iPSC-derived cells further extends this framework by enabling donor-specific modelling of arterial SMC heterogeneity and disease-relevant responses [200,201]. To enable cross-study compatibility, recent calls for unified metrics in vascular organ-on-chip research, including flow magnitude, ECM composition, and readout standardisation, should be addressed in future studies [168].

### 4.6. Mechanical Stimulation Models

Mechanical cues are primary regulators of SMC plasticity, influencing key mechanotransduction pathways such as SRF–MYOCD–MRTF and YAP/TAZ. Shear flows, either laminar or disturbed (oscillatory) flow, further modulate endothelial activation to elicit secondary paracrine signals that drive SMC phenotypic changes. Modelling these mechanical inputs allows the investigation of how stiffness, cyclic stretch, and shear converge to regulate SMC behaviour and phenotype.

#### 4.6.1. Substrate Stiffness and Topology

While the influence of matrix stiffness on SMC phenotype in 3D models was discussed in Section 4.5.2, stiffness-controlled 2D systems allow the interrogation of mechanotransduction pathways that translate mechanical tension into transcriptional reprogramming. SMCs cultured on substrates mimicking the medial environment, such as polyacrylamide or PEG substrates (∼5–10 kPa), exhibit enhanced contractile marker expression and reduced proliferation. In contrast, stiffer, plaque-resembling matrices (≥50 kPa) were shown to induce MRTFA and YAP/TAZ nuclear translocation, promoting the activation of synthetic and inflammatory gene programmes [202,203]. Cell shape control and cytoskeletal alignment have also been shown to enhance Ca^2+^ signalling synchrony, reinforcing actin-myosin organisation, to support the maintenance of contractile function [204,205]. Integrative assessment of stiffness gradients and cell topography revealed synergistic regulation of SMC elongation, proliferation, and maintenance of the contractile phenotype [206].

#### 4.6.2. Cyclic Stretch and Pressure

Models that incorporate cyclic strain aim to replicate the pulsatile deformation experienced by SMCs in vivo. In these systems, uniaxial or biaxial stretch (5–15% at 0.5–1 Hz) is usually applied to SMCs. Studies demonstrated that physiologically relevant strain enhances contractile marker expression and promotes cytoskeletal organisation. In contrast, supraphysiological stretch (>15%) enhances proliferation, ECM remodelling and inflammatory signalling [16,207]. Combining mechanical and biochemical cues shows a synergistic regulatory effect on SMC plasticity; TGF-β addition with physiological stretch promotes a fibromyocyte-like phenotype, whilst PDGF-BB stimulation under excessive strain induces a migratory synthetic SMC state [207,208].

#### 4.6.3. Shear and Transmural Flow

Adding a shear stress component to in vitro modelling enables the study of shear-dependent EC signalling and its subsequent effects on SMC phenotypic modulation. Exposure to laminar shear flow promotes the maintenance of the contractile phenotype through EC-derived mediators (e.g., NO), while disturbed or oscillatory shear flow enhances SMC phenotypic transition, proliferation, and ECM remodelling [209,210]. Findings from microfluidic and perfusion-based co-culture models showed that laminar shear activated ECs influence SMC plasticity through paracrine signalling, by suppressing inflammatory gene expression and promoting SMC alignment [211,212]. In contrast, disturbed shear flow induces EC dysfunction, triggering SMC synthetic transition and enhanced migration [211]. Beyond EC-mediated effects, transmural or interstitial flow has a direct impact on SMC ECM organisation and motility, often employed to reflect the mechanical gradient of the diseased arterial wall [213].

### 4.7. When Is Complexity Needed? When Is It Misleading?

Choosing the appropriate model to test a hypothesis involves aligning the complexity of the experiment with the scientific question and the required mechanistic detail. Simpler, minimal systems remain essential tools for examining causal relationships and studying factors and exposures individually to untangle responses. More intricate models, including co-culture and microphysiological systems, enable the exploration of additive and emergent effects at the multicellular level. However, increased complexity should be justified with a clear hypothesis, and findings should be benchmarked against human in vivo data, to minimise the risk of obscuring mechanisms and to enhance reproducibility. Therefore, effective model design should be guided by the hypothesis, the required mechanistic resolution, and the inclusion of only those components necessary to capture the relevant biological interactions. Importantly, all models retain value when used within their appropriate context; how they are interpreted, and the assumptions applied, must, however, be carefully scrutinised. Figure 3 illustrates the spectrum of in vitro model complexity and summarises the pragmatic decision path to model selection, while Table 3 compares the described models.

**Table 3 cells-14-01913-t003:** Comparative summary of in vitro models of SMC plasticity. Experimental systems capturing smooth muscle cell (SMC) plasticity range from reductionist monocultures to complex multicellular and biomechanical constructs. Each configuration provides complementary insights: primary and immortalised cultures enable controlled dissection of molecular mechanisms, co- and tri-cultures introduce endothelial and immune interactions, while 3D, hydrogel-based, and microfluidic models recapitulate extracellular-matrix, stiffness, and flow-dependent cues. Strengths, limitations, and typical applications are summarised to guide model selection in studies investigating SMC phenotypic modulation.

Model Type	Key Features/Rationale	Principal Strengths	Limitations	Optimal Applications	Representative Resources/ Key Studies
Primary SMC monocultures	Cultures of primary human or rodent SMCs under defined stimuli	Mechanistic clarity; precise control over variables; compatible with genetic (siRNA, CRISPR-Cas9) and pharmacological perturbations; scalable for high-throughput studies.	Rapid loss of contractile phenotype; donor variability and cost (human); species-specific divergence in lineage and signalling (rodent); absence of multicellular context.	Reductionist mechanistic studies; dissecting causal signalling pathways; screening and validation of regulators of the contractile-synthetic switch.	Chamley-Campbell et al., Physiol Rev 1979 [28]; Campbell et al., Clin Sci 1993 [148]; Chen et al., BMC Genomics 2016 [131]
Immortalised SMC lines	Continuous cell lines of rodent or human origin.	Highly reproducible; cost-effective; long-term culture; convenient for transfection and high-throughput assays.	Altered cytoskeletal organisation; low contractile marker expression; phenotypes deviate from primary cells; limited disease relevance.	Preliminary mechanistic screening; high-throughput assays when primary SMCs are unavailable.	Kennedy et al., Vasc Cell 2014 [151]; Mackenzie et al., Int J Mol Med 2011 [152]
Human iPSC-derived SMCs	Differentiation from iPSCs along mesodermal, neural-crest, or epicardial lineages to generate renewable organotypic human SMCs.	Human and patient-specific; genetically tractable; scalable; can capture genetic determinants of plasticity (e.g., Marfan, SVAS); compatible with organoid and microfluidic models.	Immature/foetal-like phenotype; incomplete cytoskeletal organisation; variable purity and lineage bias; maturity-dependent interpretation required.	Genetic and mechanistic studies; modelling patient-specific variation; integration into 3D, vascular-organ-on-chip, and tri-culture systems.	Kwartler et al., ATVB 2024 [154]; Cheung et al., Nat Protoc 2014 [156]; Wanjare et al., Cardiovasc Res 2013 [163]
EC–SMC co-cultures	Endothelial–SMC systems arranged in direct contact, Transwell, or conditioned-media configurations.	Preserve EC–SMC crosstalk via Notch/Jagged, PDGF, TGF-β, ET-1; maintain contractile phenotype under contact; model endothelial dysfunction and fibrous-cap regulation.	Media incompatibility; donor-matching complexity; membrane composition and pore size limit fidelity; limited longevity.	Investigating EC-driven modulation of SMC state; studying endothelial activation and paracrine vs. contact-dependent cues.	Truskey et al., Int J High Throughput Screen 2010 [171]; Liu et al., Circ Res 2009 [172]; Abbott et al., Journal of Vascular Surgery 1993 [134]
SMC–immune cell co-cultures	Bi-cultures with macrophages or T-cells in contact, insert-separated, or conditioned-media formats.	Enable study of inflammatory cues driving macrophage-like and osteogenic-like SMC states; replicate paracrine vs. juxtacrine regulation; align with plaque-derived signatures.	Lack endothelial barrier; immune-cell polarisation drift; use of immortalised lines (e.g., THP-1) may misrepresent primary cells; ratio imbalance can exaggerate effects.	Mechanistic dissection of inflammation-induced plasticity; validating cytokine- and lipid-driven transitions observed in vivo.	Weinert et al., Cardiovasc Res 2012 [181]; Schäfer et al., ATVB 2024 [185]; Deuell et al., J Vasc Res 2012 [188]
EC–SMC–immune cell tri-cultures	Three-cell Transwell or layered systems allowing endothelial activation, leukocyte transmigration, and SMC response.	Closest 2D analogue of plaque microenvironment; reproduces adhesion, barrier disruption, transmigration; recapitulates combined EC + immune cell cues inducing strong SMC phenotypic shifts.	Complex setup; ratio and insert variability; media matching and polarisation control critical; low throughput.	Modelling endothelial activation–immune cascade–SMC response axis; assessing multi-cellular inflammatory regulation of SMC fate.	Liu et al., PLOS ONE 2023 [143]; Wiejak et al., Scientific Reports 2023 [144]; Noonan et al., Front Immunol 2019 [145]
3D spheroids/aggregates	Self-assembling SMC or mixed-cell clusters forming oxygen and matrix gradients.	Generate hypoxic cores and matrix gradients; enable contractility assays; support drug perturbation screening; reflect HIF-1α/RUNX2-driven osteogenic programmes.	Core-rim heterogeneity; undefined ECM composition on non-adherent substrates; diffusion constraints.	Studying hypoxia-dependent reprogramming, calcification-related transitions, and contractile function.	San Sebastián-Jaraba et al., Clin Investig Arterioscler 2024 [191]; da Silva Feltran et al., Exp Cell Res 2024 [192]; Garg et al., Cells 2023 [193]
Hydrogel-embedded constructs	SMCs or EC–SMC mixtures embedded in defined ECM hydrogels (e.g., PEG-fibrinogen, GelMA/alginate).	Tunable stiffness and composition; support mechanotransduction analysis; demonstrate stiffness-dependent phenotype (RhoA activation, MYOCD/MRTF response).	Gel composition, crosslinking, and stiffness strongly affect outcomes; limited scalability.	Testing stiffness- and matrix-composition-dependent regulation of SMC phenotype and viability.	Stegemann et al., J Appl Physiol 2005 [195]; Peyton et al., Biomaterials 2008 [196]; Xuan et al., Tissue Eng Regen Med 2023 [197]
Tissue-engineered vascular constructs	3D rings/tubes with circumferential SMC alignment and optional EC/fibroblast layers.	Recapitulate geometry and flow; allow contractility testing (e.g., KCl-induced response); suitable for patient-specific iPSC-SMCs.	Technically demanding; requires bioreactors; low throughput.	Modelling flow and pressure effects on contractility and matrix deposition; translational disease modelling.	Dash et al., Stem Cell Reports 2016 [165]; Liu et al., Microsystems & Nanoeng 2025 [198]
Microfluidic “vessel-on-chip” systems	EC–SMC co-culture channels under laminar or disturbed flow in defined 3D ECM.	Real-time observation of EC activation and paracrine SMC modulation; captures shear-dependent phenotype switching; compatible with iPSC-derived cells.	Complex fabrication; lack of standardised metrics; limited duration.	Studying shear- and flow-dependent EC–SMC signalling; donor-specific vascular responses.	Vila Cuenca et al., Stem Cell Reports 2021 [170]; van Engeland et al., Lab Chip 2018 [199]; Liu et al., Lab on a Chip, 2023 [201]
Mechanical stimulation models (stretch, pressure, shear)	Systems applying cyclic strain (5–15%, 0.5–1 Hz), pressure, or shear flow to cultured SMCs or EC–SMC assemblies.	Quantitative control of mechanical cues; replicate pulsatile and disturbed hemodynamics; reveal activation of SRF–MYOCD/MRTF and YAP/TAZ pathways.	Require specialised equipment; simplified cellular context.	Mechanistic interrogation of mechanotransduction; coupling of mechanical and biochemical cues.	Tsai et al., Circ Res 2009 [209]; Kona et al., Open Biomed Eng J 2009 [207]

## 5. Benchmarking Models Against Human Data

### 5.1. The Need for Human-Relevant Benchmarking

Vascular biology, particularly SMC research, has historically relied on murine lineage-tracing studies and simplistic 2D cultures lacking standardisation. Whilst these approaches have yielded invaluable mechanistic insights, such as *Klf4*, *Myh11* and *Tcf21* knockouts, findings often diverge from human plaque biology [10,18]. This translational gap is not unique to vascular research; it reflects a broader issue in biomedical research, in which murine transcriptional and translational responses poorly mirror human disease [214,215]. Species differences have been covered in detail in previous sections (Section 1.3 and Section 2.3). This section discusses how modelling can be improved by systematically benchmarking findings to human data. Despite extensive preclinical work, few mechanisms validated in static cell or animal models have translated into effective therapies, underscoring the need to benchmark models directly against human data.

For decades, model systems were judged by internal consistency, whether a phenotype was reproduced in mice and in vitro, rather than by whether they accurately recapitulate a human plaque state. Early mouse lineage-tracing studies revealed that SMC phenotypic modulation occurs, but did not confirm whether an analogous transition occurs in humans [10]. Human single-cell studies later showed partial overlap, but more importantly, key distinctions, such as the TCF21 fibromyocyte state [18]. Furthermore, key differences in mixed-identity and inflammatory SMC states were reinforced between human lesions that do not neatly map to murine lineages [42,48]. Beyond differences in research methodologies, sources of divergence may be attributed to genetic background, plaque environment, such as the lack of human-relevant lipid, immune cell diversity, mechanical cues, and differences in temporal dynamics, comparing decades-long human disease progression to accelerated murine disease [216]. There is a growing recognition and calls for reform urging validation pipelines anchored in human data, promoting benchmarking to align in vitro phenotypes with in vivo human transcriptomes, proteomes, and histology [23,217,218].

This reform is further supported by the increasing policy shift, especially the FDA Modernisation Act 2.0, which eliminated the legal requirement for animal testing before human clinical trials, signifying a move towards a human-relevant focus [20,21,22]. Furthermore, programmes such as the EU’s PARC framework and the NIH/NCATS Tissue Chip initiative aim to advance organ- and vessel-on-chip models [219,220]. These systems provide quantifiable transcriptomic, mechanical and morphological endpoints that enable iterative validation against human omics data [221]. Future progress does not depend solely on developing more complex models, but also on ensuring that existing and emerging systems accurately replicate the molecular and functional signatures of human atherosclerotic disease.

### 5.2. Role of Human Single-Cell Transcriptomics

As discussed in Section 2, the growing number of single-cell and single-nucleus RNA-seq studies has transformed our understanding of SMC phenotypic diversity in human atherosclerotic disease, providing direct molecular evidence of cell-state heterogeneity. These datasets provide a gold-standard quantitative framework to benchmark experimental in vitro and in vivo models that recapitulate true human disease-relevant SMC states. Benchmarking in vitro and animal outputs against human transcriptomics ensures translational alignment. Foundational transcriptomic studies describe discrete SMC-derived phenotypes (e.g., fibromyocyte, intermediate, and inflammatory modulated), taken together, providing reference points for benchmarking [18,42,47,48]. Recent advancements in the generation of integrated plaque atlases enable cross-cohort and species harmonisation, as well as label transfer, allowing for quantitative fidelity testing [50,222].

Newer, integrative resources enhance benchmarking capacity through cross-cohort and cross-species consensus, providing a broader reference context. Traeuble et al. created an integrated human plaque atlas combining all publicly available human single-cell and single-nucleus RNA-seq data from over 259 k cells across carotid, coronary, and femoral arteries. This atlas provides a single consensus reference with validated annotations, reduced cohort-specific bias, serving as a reference-mapping tool with vessel-bed-specific views [222]. The meta-analysis by Mosquera et al. integrated over 100 k cells across multiple human studies to identify convergent vascular cell programmes and robust marker sets that validate across cohorts, enabling module scoring and pseudobulk comparison in model datasets [50].

These datasets provide a rigorous platform for cross-model validation of both single-cell and bulk outputs. Model fidelity can be evaluated through reference mapping, assigning clusters to human plaque, and module scoring of key transcriptional programmes based on validated SMC remodelling signatures. Trajectory analysis can further determine whether a modelled transition accurately represents the directionality and state progression observed in human plaques. The integrated atlas framework mitigates donor, batch, and arterial-bed biases, an invaluable advantage over a single-cohort dataset and a reproducible standard for benchmarking.

Caveats remain an important consideration when utilising single-cell datasets, including technical factors such as dissociation biases, ambient RNA and doublets. Atherosclerosis sampling bias is a limiting factor in interpretations, as most human datasets are produced from late-stage, advanced lesions, often missing early transitional states, limiting trajectory interpretations.

### 5.3. GWAS and Genetic Risk as Tools for Model Prioritisation

Human genetics provides a foundation for pinpointing the mechanisms most pertinent to disease. In CAD, large GWAS have identified hundreds of loci associated with disease risk [223,224]. Integration of these genetic variants with regulatory and single-cell maps allows the connection of variants to enhancer and other regulatory regions, target genes, and ultimately specific SMC phenotypic states. This framework allows prioritisation of disease-relevant pathways and SMC states, rather than relying on arbitrary perturbations.

Enrichment and partitioned heritability analyses demonstrate that CAD risk variants are concentrated within regulatory elements active in ECs and SMCs, especially those associated with modulated states, supporting the initiative for SMC-relevant prioritisation [225,226]. Furthermore, colocalization of CAD signals with expression quantitative trait loci (eQTLs) in human CASMCs has identified putative effector genes, including *TCF21*, *SMAD3* and *PDGFRA* [227,228]. Integration of CAD GWAS loci with multi-omic datasets, such as large-scale variant-to-enhancer-to-gene frameworks and cross-cohort meta-annotations, has enabled the refinement of variant-gene connections and the prioritisation of key gene-regulatory networks for benchmarking, providing probabilistic rather than definitive variant–gene links, which still require perturbation validation [229,230,231].

Several canonical CAD GWAS loci have contributed to the understanding of SMC phenotypic plasticity and informed priorities for model validation. The *TCF21* locus has been shown to promote the fibromyocyte transition [19,97]. The *KLF4* locus promotes SMC dedifferentiation to the LGALS3^+^ modulated state, highlighting it as a key translational target [10]. CAD-associated variation at *SMAD3*, reducing its expression in SMCs, and promoting a fate shift towards a synthetic and inflammatory state, highlights the importance of TGF-β/SMAD signalling in SMC phenotypic regulation [232,233]. Furthermore, risk variants at *PDGFD* drive increased expression and secretion of PDGFD, enhancing proliferative and migratory functions of SMCs, validating the PDGF-BB context when assessing pathological activation of SMC phenotypic modulation [234]. Together, these loci exemplify human genetics that in vivo and in vitro models should seek to recapitulate.

Integrative genomic resources, including large GWAS, single-cell regulatory maps, and gene-regulatory network models, have made CAD genetics more accessible and actionable for model benchmarking. Large CAD GWAS meta-analyses provide high-confidence variant lists and cross-ancestry replication [223,224]. Single-cell regulatory maps link risk variants to cell-specific enhancer-gene pairs [235]. New cross-trait & multi-omic analyses extend variant-to-function annotation in SMCs and support new cross-phenotype and regulatory annotation of CAD loci [231].

Some considerations to keep in mind when utilising GWAS data include distinguishing between cell type and state when mapping variant effects, to ensure separation of risk enrichment in state-specific contexts rather than lineage-wide. To ensure confidence in the analysis, co-localisation and perturbation evidence remains essential.

### 5.4. Beyond Transcriptomics: Multi-Omic Integration

While transcriptomic analysis provides valuable insights and resolution, it only captures one aspect of the cell state, which is mRNA abundance. Disease mechanisms often involve multiple layers, including epigenetic regulation, post-transcriptional and post-translational modifications that RNA alone cannot reveal. Integrating epigenomics, chromatin accessibility, proteomics, and spatial context enables a more comprehensive mapping of the SMC phenotypic diversity and plaque architecture. Integrating multi-omic data analysis and validation improves the likelihood of accurately capturing the full in vivo picture [217,236].

Epigenomic profiling provides a critical layer of information, defining the regulatory architecture of SMC plasticity. Single-cell chromatin accessibility maps (scATAC-seq) revealed enhancer landscapes and TF networks underlying the SMC phenotypic state transitions in human plaques. Aherrahrou et al. have characterised KLF4, TCF21, JUN and TEAD motifs to have distinct enhancer usage and TF activity in the modulated SMC states [230]. Depuydt et al. integrated scRNA-seq and scATAC-seq to connect transcriptional and chromatin programmes, demonstrating that chromatin accessibility correlates with SMC transition trajectories [48]. Turner et al. mapped chromatin accessibility across human CAs, identifying SMC-specific regulatory elements enriched for CAD risk variants [237]. Amrute et al. extended this foundational work by linking CAD variants to enhancer gene-pairs [235]. These comprehensive studies substantially expand our understanding of the regulatory mechanisms governing SMC phenotypic plasticity, providing chromatin-level benchmarks for model validation.

Proteomic layers are essential to validate whether transcriptional programmes translate to protein-level changes, informing post-transcriptional and post-translational modifications. Cellular Indexing of Transcriptomes and Epitopes by Sequencing (CITE-seq) and multi-omic analysis of human carotid atherosclerotic plaques identified seven distinct SMC subpopulations [238]. The extensive mass spectrometry-based proteomic analysis of human atherosclerotic lesions linking protein profiles to histology revealed plaque inflammation and calcification signatures [239]. While the ECM-focused workflows resolve ECM remodelling central to the fibrous cap, revealing that proteins involved in inflammation, ECM remodelling and protein degradation were enriched in unstable plaques [240], large cohort analyses highlight the divergence in protein and RNA abundance, reiterating the need to benchmark on the protein level to increase fidelity [241].

While dissociative analyses provide extensive molecular detail, they lack the spatial information of cell states within the atherosclerotic lesions. Spatial transcriptomics enables the uncovering of the structural organisation, anchoring SMC phenotypes within the native plaques, allowing the benchmarking of both on molecular and positional levels. Spatial transcriptomic mapping of human carotid plaques revealed SMC-derived cell niches localised to defined plaque zones [242]. In human coronary arteries, distinct SMC clusters were demonstrated to localise across plaque, medial and adventitial layers, correlating SMC state with plaque architecture [243]. A recent spatial proteomic study of human atherosclerotic plaques revealed that transcriptionally defined SMC-derived clusters localise within distinct micro-domains of the lesion, co-positioned with immune-cell-rich regions [244]. These studies demonstrate a further validation step in model benchmarking, localising certain SMC phenotypic transitions to specific regions of the atherosclerotic lesion, to ensure that the relevant processes and underlying mechanisms are recapitulated in vitro.

Multi-omic data enables the connecting of genetics with phenotype, linking variants to enhancers, genes, proteins, and downstream protein functions, with spatial context (Table 4). Multi-layer model benchmarking ensures greater accuracy of predictive relevance to human disease. Whilst this extensive integrative benchmarking is the next step in ensuring translation, challenges of multi-omic data analysis include data sparsity, batch effects across donors and platforms, and modality alignment with different dynamic ranges.

**Table 4 cells-14-01913-t004:** Representative human omics datasets informing SMC phenotypic modulation. Human genomic, transcriptomic, epigenomic, proteomic, and spatial datasets defining the molecular and cellular architecture of atherosclerotic lesions. These resources provide quantitative benchmarks for assessing the fidelity of experimental models to human smooth muscle cell (SMC) phenotypes. Together, they outline complementary layers of regulation, including genetic susceptibility (GWAS/eQTL), chromatin accessibility (scATAC-seq), transcriptional and proteomic states (scRNA-seq/CITE-seq), and spatial organisation (spatial omics), that collectively enable model validation against human disease-relevant SMC transitions.

Dataset Type	Representative Resources/ Key Studies	Principal Insights into SMC Phenotypes	Translational/Benchmarking Relevance	Notes/Caveats
Single-cell and single-nucleus RNA-seq (scRNA-seq/snRNA-seq)	Wirka et al., Nat Med 2019 [18]; Pan et al., Circulation 2020 [42]; Alencar et al., Circulation 2020 [47]; Depuydt et al., Circ Res 2022 [48]	Define discrete human SMC-derived phenotypes (fibromyocyte, intermediate, inflammatory, osteogenic-like); reveal vascular-bed-specific heterogeneity and transcriptional trajectories of modulation.	Provide a gold-standard framework for benchmarking model fidelity through label transfer, module scoring, and trajectory alignment.	Susceptible to dissociation and sampling bias; dominated by late-stage lesions; limited capture of early transitions.
Integrated single-cell plaque atlases	Traeuble et al., Nat Commun 2025 [222]; Mosquera et al., Cell Reports 2023 [50]	Combine >250 k cells across carotid, coronary, and femoral arteries to generate harmonised annotations of vascular cell states.	Enable cross-cohort and cross-species reference mapping; mitigate donor and arterial-bed bias for quantitative benchmarking.	Integration may blur cohort-specific nuances; limited spatial resolution.
Genome-wide association studies (GWAS) and eQTL integration	Aragam et al., Nat Genet 2022 [224]; Kessler et al., JACC Basic Transl Sci 2021 [223]; van der Harst et al., Circ Res 2018 [245]	Identify CAD risk loci enriched in SMC-active enhancers; link variants to genes (*TCF21*, *KLF4*, *SMAD3*, *PDGFD*) that regulate SMC state transitions.	Support genetic prioritisation of pathways most relevant to human disease; guide target selection for perturbation studies.	Require colocalisation and perturbation validation; limited cell-state resolution.
Single-cell chromatin accessibility (scATAC-seq)	Aherrahrou et al., Circ Res 2023 [230]; Depuydt et al., Cir c Res 2022 [48]; Turner et al., Nat Genet 2022 [237]; Amrute et al., medRxiv 2024 [235]	Map enhancer landscapes and transcription-factor motif activity (e.g., *KLF4*, *TCF21*, *JUN*, *TEAD*), defining regulatory networks underlying SMC transitions.	Provide chromatin-level benchmarks connecting CAD variants to active enhancers and transcriptional programmes.	Sparse coverage per cell; donor and arterial heterogeneity; require integration with transcriptomic data.
Proteomic and CITE-seq datasets	Bashore et al., ATVB 2024 [238]; Theofilatos et al., Circ Res 2023 [239]; Lorentzen et al., Matrix Biol Plus 2024 [240]; Palm et al., Cardiovasc Res 2025 [241]	Identify post-transcriptional divergence; define SMC-derived subpopulations by surface proteins; link ECM remodelling, inflammation, and calcification to protein abundance.	Validate whether transcriptional programmes translate to protein-level changes; benchmark translational fidelity of models.	Lower proteomic depth than RNA; antibody and detection bias; RNA–protein discordance common.
Spatial transcriptomics and proteomics	Pauli et al., bioRxiv 2025 [242]; Campos et al., EMBO Mol Med 2025 [243]; Jokumsen et al., bioRxiv 2025 [244]	Localise transcriptionally defined SMC phenotypes within plaques; map zonation across intima, media, and adventitia; identify EC–immune–SMC interfaces.	Enable spatial benchmarking linking molecular states to anatomical regions; integrate morphology with molecular validation.	Resolution remains limited; most data derive from late-stage human lesions; high analytical cost and complexity.

## 6. Implications for Modelling Disease and Therapeutic Discovery

### 6.1. Importance of Model Transparency and Reproducibility

Advances in single-cell and multi-omic technologies are transforming our understanding of SMC diversity within atherosclerotic lesions. The translational potential of these insights for effective cardiovascular treatments depends on the accuracy, reliability, and transparency of disease model systems. Imperfect modelling has historically hampered therapeutic progress. Combining past translational gaps with new multi-omic frameworks will be crucial for establishing predictive, reproducible, and clinically relevant models.

To ensure that findings from in vitro and in vivo systems are comparable across studies, transparency must be maintained in experimental design, data collection and processing, and validation. Details of cell sources, including donor variability, passage number, and culture and experimental conditions, alongside data analysis pipelines, should be reported [246]. In the case of large datasets, open data and code sharing, aligned with public repositories such as Gene Expression Omnibus, ArrayExpress, and HuBMAP, enhances transparency and the utilisation of existing datasets [247]. Implementing such transparent and reproducible practices will improve the reliability of preclinical findings and strengthen their translational potential.

### 6.2. Therapeutic Implications of Poorly Modelled States

Historically, several therapeutic strategies failed in clinical translation, partly due to the underrepresentation of cellular diversity in preclinical models, the lack of mechanical context, and inflammatory feedback. For example, the CANTOS trial confirmed that IL-1β blockade reduced major cardiovascular events, but did not eliminate residual inflammatory risk and increased infections, highlighting the pathway complexity not captured in simplified model systems [133]. Several therapies aimed at raising HDL, designed to block the cholesteryl ester transfer protein (CETP), increased HDL levels, without improving outcomes [248]. Worse still, torcetrapib caused increased cardiovascular events and mortality, and off-target blood pressure events [249]. These failed therapies demonstrated that circulating HDL-cholesterol levels as a surrogate endpoint capture only one layer of disease biology and may not reflect the cellular and structural determinants of plaque stability observed in human lesions. Despite the strong preclinical evidence and rationale, broad-spectrum MMP inhibition has not shown clinical benefit for atherosclerosis and development was limited by off-target musculoskeletal toxicity and poor selectivity [250]. Systemic PDGFR inhibition with oral imatinib showed no benefit in a randomised trial of restenosis prevention, despite encouraging preclinical data [251]. In contrast, anti-proliferative drug-eluting stents (DES) targeting SMC proliferation have successfully reduced restenosis, demonstrating the value of spatially confined SMC modulation, yet can delay endothelial recovery, reinforcing the need for models that capture EC–SMC–ECM dynamics [252,253].

### 6.3. The Future: Integrating Omics and Model Systems

Together, the limitations of past modelling and therapeutic efforts highlight the need for a paradigm shift in disease modelling. Recent multi-omic and spatial frameworks provide an unprecedented reference for human atherosclerotic insights, revealing the regulatory networks and cellular transitions that drive SMC plasticity. The next step is to translate these datasets into predictive experimental systems capable of replicating the phenotypic diversity, mechanical cues and microenvironmental complexity of the plaque environment.

Integrating the wealth of knowledge into model development will guide the refinement of culture conditions, matrix composition, and stimulus design, ensuring that experimental perturbations accurately mimic disease-relevant states. Future SMC models combining multi-omic benchmarking, mechanical conditioning, and dynamic cellular interactions central to lesion progression will enable predictive vascular modelling, accelerating the potential to identify causal mechanisms and therapeutic targets in atherosclerosis.

## 7. Conclusions

SMC plasticity underlies the core of atherosclerotic disease progression, yet much of our understanding remains based on murine systems that only partially capture the phenotypic diversity of human SMCs. The growing number of human single-cell, transcriptomic, proteomic, and spatial datasets enables benchmarking models against human disease states. Integrating these omic frameworks for benchmarking will be vital to reproduce the environmental and multicellular cues that drive SMC phenotypic transitions in vitro, thereby improving data reproducibility and model accuracy. Aligning preclinical model development with human vascular biology offers the most promising pathway to bridge the translational gap, refine our understanding of mechanisms governing SMC phenotypic plasticity, and accelerate therapeutic discoveries for atherosclerotic disease. 

## Figures and Tables

**Figure 1 cells-14-01913-f001:**
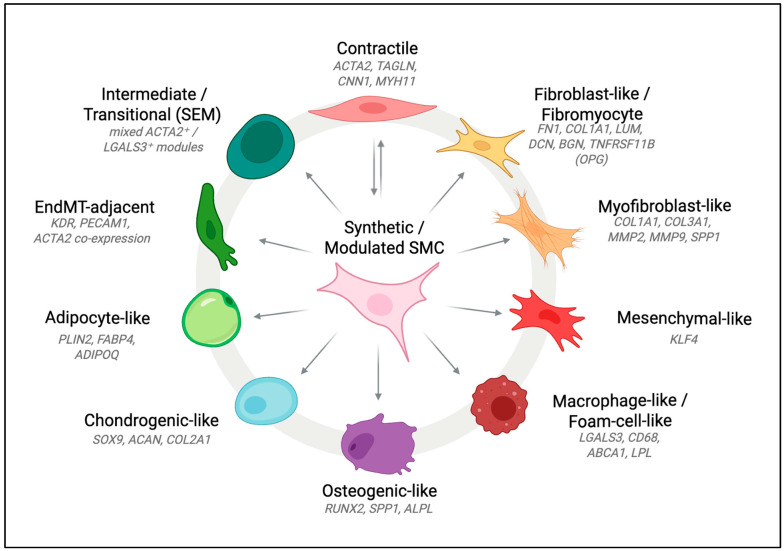
Schematic illustrating the spectrum of smooth muscle cell (SMC) phenotypic states observed in atherosclerosis. The central synthetic/modulated SMC represents a transitional hub from which distinct phenotypes emerge, including contractile, fibroblast-like, myofibroblast-like, mesenchymal, macrophage-like, osteogenic, chondrogenic, adipocyte-like, intermediate/transitional (stem cells, endothelial cells, monocytes; SEM), and endothelial to mesenchymal transition (EndMT)-adjacent states. Representative molecular markers are shown for each phenotype. Arrows indicate phenotypic interconversion, reflecting the continuum of SMC plasticity.

**Figure 2 cells-14-01913-f002:**
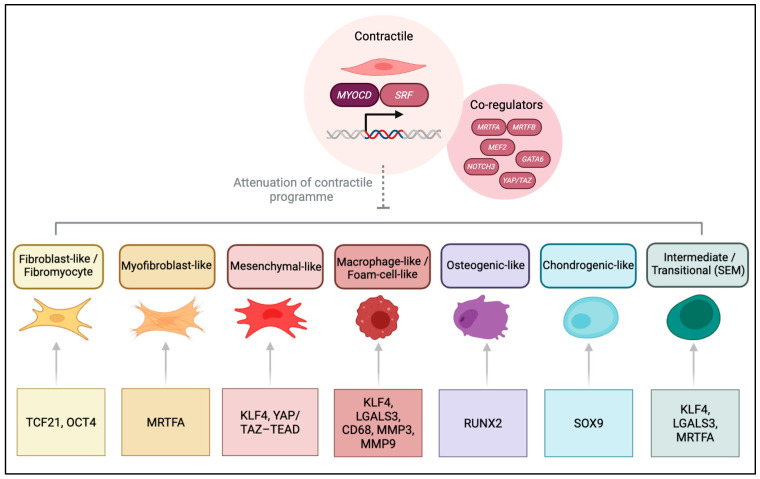
Transcriptional regulation of smooth muscle cell (SMC) phenotypic transitions. The SRF–MYOCD programme maintains the contractile SMC phenotype by activating the expression of canonical markers (*ACTA2*, *CNN1*, *TAGLN*, *MYH11*). Co-regulators, including MRTF-A/B, MEF2, GATA6, NOTCH3 and YAP/TAZ–TEAD support this programme by integrating mechanical and developmental cues. Context-specific transcriptional regulators then promote transitions toward distinct synthetic states: TCF21 and OCT4 (fibroblast-like/fibromyocyte), MRTF-A (myofibroblast-like), KLF4 and YAP/TAZ–TEAD (mesenchymal-like), KLF4, LGALS3, CD68, MMP3, MMP9 (macrophage-like/foam-cell-like), RUNX2 (osteogenic-like), SOX9 (chondrogenic-like), and KLF4, LGALS3, MRTF-A (intermediate/transitional). Together, these regulators redirect SMCs from the SRF–MYOCD contractile programme toward diverse phenotypes observed in atherosclerosis.

**Figure 3 cells-14-01913-f003:**
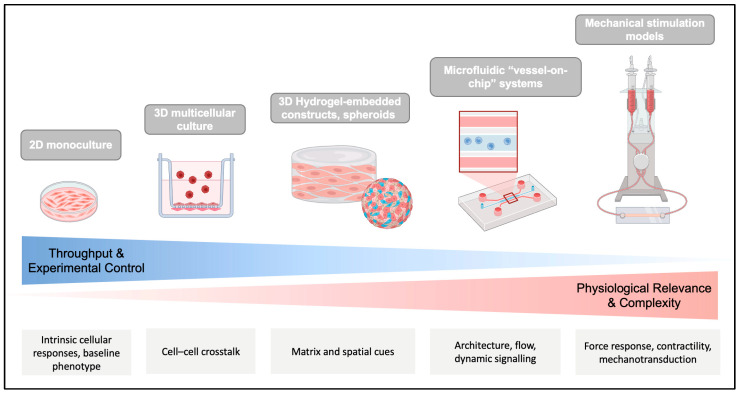
Spectrum of in vitro model complexity and relevance in the study of smooth muscle cell (SMC) plasticity. In vitro systems range from 2D monocultures, enabling high-throughput analysis of cell-autonomous responses, to 3D multicellular and matrix-based constructs that capture cell–cell and extracellular cues. Microfluidic “vessel-on-chip” systems and mechanical stimulation models integrate architectural and biomechanical parameters, achieving higher physiological relevance. Together, these models span a continuum balancing the need for experimental control with biological complexity in the investigation of SMC phenotype transitions.

## Data Availability

No new data were created or analysed in this study.

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
