# Peer review of "Bridging the Translational Gap: Rethinking Smooth Muscle Cell Plasticity in Atherosclerosis Through Human-Relevant In Vitro Models"

_cells, 2025, doi:10.3390/cells14231913_

Round 1
Reviewer 1 Report
Comments and Suggestions for Authors
This is an outstanding and much-needed review paper in the field of vascular biology, especially in light of the persistent failure of clinical trials and the high prevalence of cardiovascular disease. Although written in a measured tone, the review successfully highlights numerous critical issues that have been overlooked due to the longstanding reductionist and non-translational approaches dominating the field. The authors skillfully synthesize the most important studies and provide a thoughtful, critical analysis.
There are, however, several aspects that the authors should consider to further strengthen the manuscript. These revisions would help the review become more incisive and potentially transformative:
- The observed rhomboid morphology of cells in culture is likely an in vitro artifact and should not be interpreted as evidence of a similar phenotype in vivo.
- The introduction is overly extensive; a more concise and direct framing would improve readability.
- Simple monocultures invariably begin with dedifferentiated cells, which is a limitation, since contractile vascular smooth muscle cells are not proliferative.
- The absence of reliable surface markers remains a major challenge in defining vascular smooth muscle cell phenotypes without differentiation and to facilitate sorting.
- Eliminating fibroblast contamination from primary SMC cultures is virtually impossible; this limitation should be acknowledged.
- The review does not mention the critical role of heparin in maintaining the contractile phenotype in vitro.
- The concept of irreversibility across phenotypic transitions is not fully addressed and deserves deeper discussion.
- The authors should note that rodent vascular cells proliferate faster and more robustly in culture than human cell lines, which affects interpretation.
- The review does not clearly differentiate across vascular beds. Veins and arteries differ markedly not only in histological organization but also in cellular composition. Because the review focuses on arterial biology, this should be clearly stated, and broad generalizations should be avoided.
- The review does not sufficiently consider the possibility of artifactual mouse science. Rodent models may share little relevance to human arterial disease, and many observations may reflect artifacts, particularly given that mice do not spontaneously develop arterial pathology. The manuscript still gives these models too much benefit of the doubt despite the proven lack of translational value of mouse models.
- The chronic nature of cardiovascular disease suggests that plaque development may arise through multiple mechanisms over time, potentially making meaningful prevention of initial deposition with age unrealistic. The remarkable variability of human plaque histology supports this idea. Unlike in animals, no two human plaques are identical. This should be highlight.
- Include side-by-side pictures of human and mouse plaque. This will illustrate the differences.
- The authors should consider bioreactors using ex vivo human vascular tissues as an alternative model for studying acute vascular responses.
- Finally: all models are useful; it is the interpretation that must be scrutinized.
Reviewer 2 Report
Comments and Suggestions for Authors
This is an interesting and timely review that covers an important topic where there is considerable research interest. The authors nicely and comprehensively demonstrate a deep understanding of the complex challenges posed by a huge translational gap in our understanding of disease-relevant cell states (specifically SMC plasticity and phenotypic modulation) in atherosclerosis and present cutting-edge solutions to mitigate and circumvent the challenges we currently facing. Particularly, they proposed to benchmark data generated from in vitro or in vivo SMC phenotypic modulation studies against single-cell and multi-omics data from human atherosclerosis to further refine and validate the model systems used in future studies.
I only have several minor suggestions for them to consider as detailed below (mainly regarding the referencing in tables):
Specific comments/suggestions:
- Based on the figure legend and our current understanding about SMC phenotype switching in AS, it would be more appropriate if a reciprocal arrow is used between contractile and the central synthetic/modulated SMC in Figure 1.
- The actual reference number should be given for the representative studies in Table 1, which allow the readers to easily identify each study for more details (similar to the reference citation within the main text).
- The key references should be included in Table 2 to support their key statement regarding the functions/implication of each regulator in SMC plasticity.
- Similarly, the key references should be included in Table 3, allowing the readers to trace back the original publications/references for further details for each model.
- The actual reference number should be given for the representative key studies in Table 4, which allow the readers to easily identify each study for more details (similar to the reference citation within the main text).
- Moreover, it appears there is a typo in Table 4: ‘Mosquera et al., Cell Genom2023’ should be ‘Mosquera et al., Cell Report 2023’. Please correct it.
